# Multi-body SE(3) Equivariance for Unsupervised Rigid Segmentation and Motion Estimation

**Jia-Xing Zhong, Ta-Ying Cheng, Yuhang He, Kai Lu, Kaichen Zhou**✉,
**Andrew Markham, Niki Trigoni**
Department of Computer Science, University of Oxford
{jiaxing.zhong, ta-ying.cheng, yuhang.he, kai.lu, rui.zhou}@cs.ox.ac.uk
{andrew.markham, niki.trigoni}@cs.ox.ac.uk

## Abstract

A truly generalizable approach to rigid segmentation and motion estimation is fundamental to 3D understanding of articulated objects and moving scenes. In view of the closely intertwined relationship between segmentation and motion estimates, we present an SE(3) equivariant architecture and a training strategy to tackle this task in an unsupervised manner. Our architecture is composed of two interconnected, lightweight heads. These heads predict segmentation masks using point-level invariant features and estimate motion from SE(3) equivariant features, all without the need for category information. Our training strategy is unified and can be implemented online, which jointly optimizes the predicted segmentation and motion by leveraging the interrelationships among scene flow, segmentation mask, and rigid transformations. We conduct experiments on four datasets to demonstrate the superiority of our method. The results show that our method excels in both model performance and computational efficiency, with only 0.25M parameters and 0.92G FLOPs. To the best of our knowledge, this is the first work designed for category-agnostic part-level SE(3) equivariance in dynamic point clouds.

## 1   Introduction

Comprehending point cloud motion is critical for various 3D vision tasks in dynamic scenarios, *e.g.*, tracking, animation, simulation, and manipulation. Many types of 3D motion can be described as a composition of multi-body rigid movements, such as articulated objects [68], and vehicular traffic scenes [25]. Specifically, the setting of *multi-body rigid motion* requires all moving parts (*i.e.*, bodies) to undergo only translation and rotation, without any type of deformation. The process of modeling these multi-body rigid movements typically involves two primary portions [32]: 1) the identification of distinct moving bodies (*i.e.*, *rigid segmentation*), and 2) the calculation of individual movements for each identified body (*i.e.*, *motion estimation*).

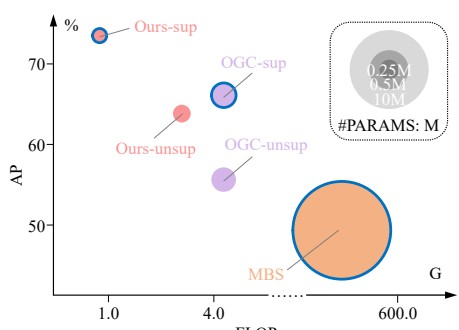

Figure 1: *Comparison of FLOPs, parameter number, and AP on SAPIEN.* Supervised methods are marked in ◯.

Being the first modeling portion, rigid segmentation significantly differs from traditional semantic segmentation tasks that concentrate on category information [16, 3, 30, 31, 70]. Based on multi-body motion instead of category-dependent semantics, rigid segmentation becomes *category-agnostic about moving parts* as rigid bodies are not limited to a specific set of shapes and naturally cannot be assigned certain category labels. As regards motion

37th Conference on Neural Information Processing Systems (NeurIPS 2023).

estimation, individual pose variations would occur arbitrarily, including transformations unseen in the training data. This constitutes an *open set of pose changes*.

Considering the relative inaccessibility and the constrained generalizability of manual annotations, Song and Yang [61] have proposed a seminal work for unsupervised rigid segmentation. Nevertheless, the unsupervised paradigm remains a formidable challenge because of the *generalizable requirement* in multi-body movements and the *interdependent nature* between rigid segmentation and motion estimation. Due to the aforementioned category-agnostic rigid prior and open-set pose changes in multi-body motion, a model is required to generalize well to transformations of appearance, location, and orientation. Moreover, rigid segmentation and multi-body motion estimation are highly coupled — the calculation of movements for each body is based on the output of a rigid partition, while an estimate of multi-body motion facilitates further segmentation. In absence of precise supervision, it is difficult to independently obtain effective training signals for either rigid segmentation or motion estimation.

To tackle the difficulties of generalizability and interdependence, this paper presents a three-fold contribution in the form of architecture, training strategy, and experimental evaluation:

1. **Architecture.** *We design a part-level SE(3)-equivariant network that demonstrates strong generalization to open-set motion and category-agnostic moving bodies.* The SE(3)-equivariance allows features of a model to keep coherence with the rigid transformations of point cloud inputs, thereby enhancing the model's robustness to such spatial transformations (*i.e.*, rigid motion) [13, 17, 9, 14, 67] and therefore is more generalizable for open-set pose changes of global rigid targets [63, 43, 8, 53]. Tailing the SE(3)-equivariance is two lightweight heads for segmentation and motion estimation. The former, unlike previous segmentation networks using global SE(3) equivariance, exhibits *point-level local flexibility* to SE(3)-invariance. The latter, robustly employing the probability of part consistency, utilizes *part-level "softly" matching operations* to handle SE(3)-equivariance without requiring knowledge of part categories. As a result of integrating *two lightweight heads* into a unified network, rather than utilizing large independent networks, our model is characterized by a small number of parameters (0.25 M) and low computational complexity (0.92G FLOPs). To the best of our knowledge, this is the first work designed for *category-agnostic part-level SE(3)-equivariance* in dynamic point clouds.

2. **Training Strategy.** *We leverage the interdependence between rigid segmentation and motion estimation and present a unified training strategy to jointly optimize their outputs in an online fashion.* Based on our proposed two-head network, we simultaneously filter out the noisy flow predictions and refine the estimates of rigid motion by exploiting the interrelation among scene flow, segmentation mask, and rigid transformation. Superior to previous arts, our online training process is *free from complicated components and manual intervention*, *e.g.*, the alternation of Markov Chain Monte Carlo proposals and Gibbs sampling updates [28], the offline repetition of training multiple segmentation networks from scratch [61], or the continual optimization of an accessory neural network of flow estimation [32].

3. **Experiments.** *Experiments on four datasets demonstrate the efficacy of our model in performance, as well as its efficiency in parameters and computational complexity.* We conduct comprehensive experiments on four datasets (SAPIEN [68], OGC-DR [61], OGC-DRSV [61], and KITTI-SF [48]) across three application scenarios (articulated objects, furniture arrangements, and vehicular traffic). Noticeably, as shown in Figure 1, our performance on the SAPIEN dataset of articulated objects surpasses state-of-the-art results w.r.t. all evaluation metrics, achieving the relative gain of at least 14.7% AP in rigid segmentation and at least 23.3% in motion estimation, with only 0.25M parameters and 0.92G FLOPs. The code is available at https://github.com/jx-zhong-for-academic-purpose/Multibody_SE3.

## 2   Related Works

**Deep Learning on Dynamic Point Clouds.**   Deep neural networks for point cloud video modeling aim to understand our dynamic 3D surroundings. These networks enable the resolution of several downstream tasks, such as video-level classification [20, 22, 44, 23, 21, 74, 29], frame-level prediction [57, 36, 50], object-level detection [6, 40, 54, 58] or tracking [26, 56, 72], part-level mobility parsing [60, 59, 71], point-level segmentation [22, 23, 7, 20], and scene flow estimation [18, 64, 34, 11, 2]. In contrast to the above dynamic tasks that usually assume contiguous

sequential input, multi-body rigid motion may be captured between discrete frames [32], presenting a unique challenge.

**3D Motion Segmentation & Multi-body Rigid Motion.** Although the systematic formulation of multi-body rigid motion modeling is a relatively recent development [32], a related problem, *i.e.*, motion segmentation, has been long sought after. Motion segmentation aims to group points with similar motion patterns from the input of scene flow, using such techniques as factorization [10, 42, 69], clustering[33], graph optimization [47, 35, 4], and deep learning [39, 28, 71, 1]. However, motion segmentation does not explicitly take into account multi-frame multi-body consistency. To address this issue, Huang *et al*. [32] present the seminal fully-supervised work for modeling multi-body rigid motion. Song and Yang [61] further attempt to predict the mask of rigid segmentation via three object geometry losses in an unsupervised setting. Following the same unsupervised paradigm, our approach has distinct motivations: 1) to achieve high generalizability in the presence of low-quality training signals, and 2) to simplify the training process through online optimization.

**SE(3) Equivariant Networks for Point Clouds.** Equivariant networks have superior discriminative and expressive ability for various data structures (*e.g.*, graphs [5, 65], images [15, 49]) under the transformation of some symmetry groups. Among them, the equivariance of a Special Euclidean group (3) (SE(3)) has drawn increasing attention to 3D point cloud processing [13, 17, 9, 14, 67, 63, 43, 8, 53]. A vast majority of these researches focus on global, while the part-level local equivariance is under-explored. Recently, [73] has utilized part-level SE(3) equivariance for supervised bounding box detection. Concurrently with our work, Lei *et al*. [41] and Feng *et al*. [24] apply part-level equivariance to category-specific tasks of object segmentation and human body parsing, respectively. Differently, we leverage part-level equivariance to the unsupervised category-agnostic problem.

# 3 Methodology

## 3.1 Background: SE(3)-equivariance/invariance & Discretization

Given a point cloud $X \in \mathbb{R}^{n \times 3}$ of $n$ points and a rigid transformation $\mathbf{T} \in \text{SE}(3) : \mathbb{R}^{n \times 3} \to \mathbb{R}^{n \times 3}$, a neural network mapping inputs to a feature domain $\phi : \mathbb{R}^{n \times 3} \to \mathcal{F}$ is defined as *SE(3)-equivariant* if $\phi(\mathbf{T} \circ X) = \mathbf{T} \circ \phi(X), \forall \mathbf{T} \in \text{SE}(3)$, where $\circ$ means performing an SE(3) transformation. Likewise, *SE(3)-invariance* is expressed as $\phi(\mathbf{T} \circ X) = \phi(X)$. To reduce the computational cost of equivariant networks, [12] discretizes the rotation space into an icosahedral group $\mathcal{G}$ of 60 rotational angles ($|\mathcal{G}| = 60$). As a further extension, Equivariant Point Network (EPN) [9] operates the SE(3) discretization on point clouds. For a point $x \in \mathbb{R}^3$ in the input $X$, the feature extractor of EPN outputs the per-point representation $f(x) \in \mathbb{R}^{|\mathcal{G}| \times D}$, where $D$ is the feature dimension. By definition, $f(x)$ is SE(3)-equivariant to its neighbors of the convolution kernel's receptive field:

1) Rotation equivariance within the icosahedral group: $f(g \circ x) = g \circ f(x), \forall g \in \mathcal{G}$.

2) Translation invariance w.r.t. arbitrary translation $t$: $f(t \circ x) = f(x)$.

By relaxing the strictly rotational equivariance into the equivariance w.r.t. the 60 icosahedral angles in $\mathcal{G}$, EPN is proven to be robust to many downstream tasks, such as 6D pose estimation [43], and place recognition [45]. Due to its high robustness, we choose EPN as our SE(3) equivariant backbone of feature extraction. In principle, our training strategy is versatile, and other networks with *per-point SE(3)-equivariant representations* can also serve as its feature extractor.

## 3.2 Problem Statement

We define a set of point clouds $P$ from $K$ frames that are not necessarily consecutive as $P = \{P_1, P_2, \cdots, P_{K-1}, P_K\}$, where each frame $P_k = \{p_k^i \in \mathbb{R}^3 | i = 1, 2, \cdots, N-1, N\}$ contains $N$ three-dimensional points. All frames of $P$ represent the same target,[1] which is segmented into $S$ independently moving rigid parts. For a given frame $P_k$, the point-part rigid mask is defined as $\mathbf{M}_k \in \{0, 1\}^{N \times S}$. $\mathbf{M}_k^{is} = 1$ indicates that the point $p_k^i$ belongs to the $s^{th}$ rigid part; $\mathbf{M}_k^{is} = 0$ indicates that it does not. The rigid motion of the $s^{th}$ part between two frames $(P_k, P_l)$ is denoted

---

[1]The term "target" refers to objects, environments, or scenes associated with multi-body rigid motion, such as articulated objects, arrangements of indoor furniture, or scenes of vehicular traffic.

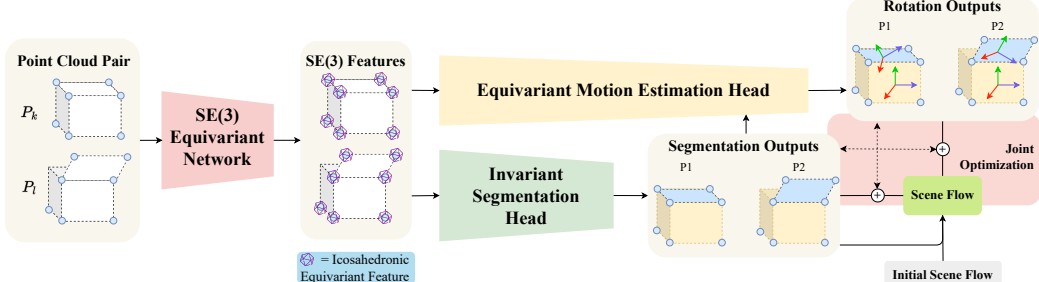

Figure 2: *An overview of network structure and training strategy.* We feed a pair of point clouds $P_k, P_l$ into the SE(3) equivariant networks to obtain a pair of SE(3) icosahedronic features. These features are then fed into the two proposed heads for motion segmentation and invariant segmentation. The two outputs, combined with scene flow, are used for a joint optimization that can be done in an unsupervised manner.

as $\mathbf{T}_{kl}^s \in \text{SE}(3)$. This rigid transformation is specified by a rotation matrix $\mathbf{R}_{kl}^s \in \text{SO}(3)$ and a translation vector $\mathbf{t}_{kl}^s \in \mathbb{R}^3$. Note that the targets in the dataset belong to various categories, including classes possibly unseen in the training data.

Given the point cloud set $P$, we aim to segment each frame $P_k$ with rigid-part masks $\mathbf{M}_k$, and obtain part-level rigid transformation $\mathbf{T}_{kl}^s$ between frames. The former task is called *rigid segmentation*, while the latter is named *motion estimation*. In the supervised setting, ground-truth segmentation masks and rigid motions (or scene flow) are provided in the training data. In the unsupervised formulation, neither segmentation nor motion information is available, and our training input solely consists of point cloud frames.

## 3.3  Network Architecture

As shown in Figure 2, our network takes a pair of point cloud frames as the input and outputs the predictions of rigid segmentation and motion estimation. There are three main components in the proposed framework, *i.e.*, a feature extractor, an invariant segmentation head, and an equivariant motion estimation head.

**Per-point Feature Extractor.**  For an input frame $P_k$, the feature extractor of EPN outputs per-point SE(3)-equivariant representations $F_k \in \mathbb{R}^{N \times |\mathcal{G}| \times D}$, following the notations in Section 3.1 and 3.2. The corresponding feature $f_k^i \in \mathbb{R}^{|\mathcal{G}| \times D}$ of a point $p_k^i$ can be viewed as a concatenation of different representations w.r.t. $g_j \in \mathcal{G}$ over the rotation group dimension:

$$f_k^i = [\theta(p_k^i, g_1), \theta(p_k^i, g_2), \cdots, \theta(p_k^i, g_{|\mathcal{G}|-1}), \theta(p_k^i, g_{|\mathcal{G}|})], \tag{1}$$

where $\theta$ is a $D$-dimensional encoder based on a stack of convolution kernels. The prediction module of vanilla EPN is designed for global SE(3)-equivariance for all input points. Differently, our unsupervised multi-body task requires the model's ability to handle *part-level local equivariance*, especially under low-quality training signals. For this purpose, we further devise two heads for rigid segmentation and motion estimation.

**Point-level Invariant Segmentation Head.**  Rigid segmentation is an SE(3)-invariant task, as the predicted mask should remain consistent for the same points across various poses and positions. Traditional SE(3)-equivariant structures assume that all input points undergo the same rigid transformation, which is not in alignment with the multi-body setting. To *encode distinct transformations for individual input points*, the equivariant features $\theta(p_k^i, g_j)$ are aggregated into an invariant representation $u_k^i$ across the dimension of the rotation group, as depicted in Figure 3:

$$u_k^i = \sum_{j}^{|\mathcal{G}|} w(p_k^i, g_j)\theta(p_k^i, g_j), \tag{2}$$

where $w(p_k^i, g_j) \in [0, 1]$ is a selection probability of the discrete rotation $g_j$ in $\mathcal{G}$, derived through a $1 \times 1$ convolution. The weighted sum $u_k^i$ is invariant to rigid motion given a point with its neighbors

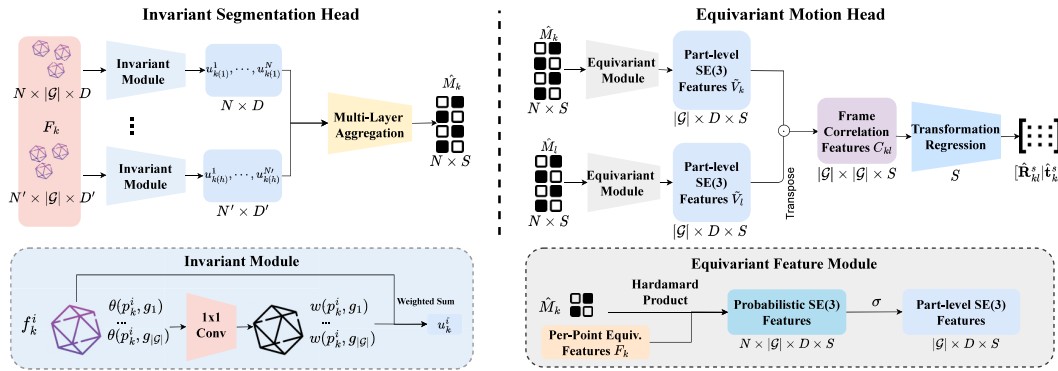

Figure 3: *An overview of Segmentation and Motion heads.* The invariant segmentation head comprises an invariant module that sums $\theta(\cdot)$ and $w(\cdot)$ to obtain an invariant representation which is then aggregated through multiple layers to obtain a segmentation mask. Pairs of segmentation masks $\hat{M}_k, \hat{M}_l$ are fed into the equivariant motion head along with per-point equivariant features $F_k$ to obtain correlations features for predicting the final transformations of points.

in the convolution receptive field. As Figure 3 illustrates, by fusing such invariant representations $u^i_{k(1)} \in \mathbb{R}^D, \cdots, u^i_{k(h)} \in \mathbb{R}^{D'}$ from $h$ layers in the feature extractor, the segmentation head outputs a soft prediction $\hat{M}_k \in [0,1]^{N \times S}$ of the rigid mask.

**Part-level Equivariant Motion Estimation Head.** Part-level SE(3)-equivariance is desirable for motion analysis, especially rotation estimation. Based on the noisy predictions $(\hat{M}_k, \hat{M}_l)$ of the frames $(k, l)$ from the head of rigid segmentation, the motion head is supposed to *handle these uncertain category-agnostic parts*. Figure 3 demonstrates the operational scheme of our motion estimation head. First of all, the part-level SE(3) feature $V_{k,j} \in \mathbb{R}^{N \times D \times S}$ w.r.t. the rotation $g_j \in \mathcal{G}$ of a single frame $k$ is obtained from the per-point equivariant representations $F_k$ and the predicted rigid mask $\hat{M}_k$:

$$V_{k,j} = \{\hat{m}^1_k \cdot \theta(p^1_k, g_j), \hat{m}^2_k \cdot \theta(p^2_k, g_j), \cdots, \hat{m}^{N-1}_k \cdot \theta(p^{N-1}_k, g_j), \hat{m}^N_k \cdot \theta(p^N_k, g_j)\}, \quad (3)$$

where $\hat{m}^i_k$ is the element corresponding to a point $p^k_i$ in $\hat{M}_k$, and $\cdot$ is the broadcast operation of Hardamard product. Afterward, $V_{k,j}$ over the rotation group $\mathcal{G}$ is concatenated as $V_k \in \mathbb{R}^{N \times |\mathcal{G}| \times D \times S}$, followed by a permutation-invariant operation $\sigma: \mathbb{R}^{N \times |\mathcal{G}| \times D \times S} \to \mathbb{R}^{|\mathcal{G}| \times D \times S}$ (*e.g.*, max pooling) over all the points to produce part-level equivariant features $\tilde{V}_k = \sigma(V_k)$. Between two frames $(k, l)$, the part-level rotation correlated feature $C_{kl} \in \mathbb{R}^{|\mathcal{G}| \times |\mathcal{G}| \times S}$ is defined as:

$$C_{kl} = \tilde{V}_k \tilde{V}_l^T, \quad (4)$$

where $^T$ is matrix transposition. $C_{kl}$ is calculated upon "softly matching" within each consistent rigid part, while the specific category labels can be agnostic to the model. Based on the correlated feature $C_{kl}$, the motion head estimates rotation $\hat{\mathbf{R}}^s_{kl}$ and translation $\hat{\mathbf{t}}^s_{kl}$ of each rigid part $s$. More implementation details can be found in **Supp**.

## 3.4 Unsupervised Training Strategy

Ideally, the relation among scene flow $\delta_{kl} \in \mathbb{R}^{N \times 3}$, rigid segmentation $M^s_k$ of the $s^{th}$ part, and multi-body transformation $\mathbf{T}^s_{kl}$ is as follows:

$$P_l = P_k + \delta_{kl} = \bigcup_s^S \{\mathbf{T}^s_{kl} \circ (M^s_k \star P_k)\}, \quad (5)$$

wherein the operation of union set is denoted as $\bigcup$, and $\star$ signifies the removal of a point $p^k_i$ if it does not in the rigid $s$. Nevertheless, their predictions $\hat{\mathbf{T}}^s_{kl}$, $\hat{M}_k$ and $\hat{\delta}_{kl}$ may exhibit a considerable amount of interdependent noise during the unsupervised training process. To *mitigate the noise in one prediction by leveraging the other two*, we propose an online training strategy without interrupting the end-to-end optimization of a network, as shown in Figure 2.

**Cold Start & Scene Flow $\hat{\delta}_{kl}$ Updating.** Scene flow creates a relation between motion and segmentation. As a dense vector field that maps points to points, it can directly assist point-level segmentation. At the same time, scene flow also contains movement information for motion estimation. We initialize by collecting noisy scene flow from an unsupervised flow estimator (*e.g.*, FlowStep3D [38]), and calculate rudimentary estimates of rigid masks and transformation by minimizing the discrepancy between their derived displacement and scene flow. As the accuracy of their estimates $\hat{M}_k^s$ and $\hat{\mathbf{T}}_{kl}^s$ improves, the scene flow is online corrected during a training epoch:

$$\hat{\delta}_{kl} = \alpha\hat{\delta}_{kl} + (1-\alpha)(\bigcup_s^S \{\hat{\mathbf{T}}_{kl}^s \circ (\hat{M}_k^s \star P_k)\} - P_k), \tag{6}$$

where $\alpha \in [0, 1]$ is a decay factor to control the updating rate. In this manner, improved scene flow is capable of providing enhanced supervision to learn segmentation masks and motion estimates.

**Motion & Flow $\rightarrow$ Segmentation $\hat{M}_k^s$.** Although scene flow encodes point-level supervisory signals for segmentation, accurately computing flow between non-adjacent frames is challenging, as pointed out by Song and Yang [61]. To alleviate the influence of miscalculated scene flow, we optimize our segmentation head based on the *consensus between the motion head and updated scene flow*. Given the outputs $\hat{\mathbf{R}}_{kl}^s, \hat{\mathbf{t}}_{kl}^s$ of our motion head, and the interpolation point $p'^i_l = p_k^i + \delta_{kl}^i$ derived from scene flow, the consensus score $\beta_{kl}^{s(i)}$ of the point $p_k^i$ w.r.t. the rigid $s$ between the frames $(k, l)$ is defined as:

$$\beta_{kl}^{s(i)} = exp(-\tau||\hat{\mathbf{R}}_{kl}^s p_k^i + \hat{\mathbf{t}}_{kl}^s - p'^i_l||), \tag{7}$$

where $\tau$ is a temperature coefficient to set the sharpness of $\beta_{kl}^{s(i)}$, and $||\ ||$ is $\ell 2$-norm. Intuitively, a low $\beta_{kl}^{s(i)}$ indicates a large disagreement between scene flow and the motion head, of which the mask should have a small learning weight in the segmentation loss:

$$l_{seg} = \frac{1}{NS} \sum_i^N \sum_s^S ||\beta_{kl}^s \hat{M}_k^s(p'^i_l - \hat{\mathbf{T}}_{kl}^s \circ p_k^i)||, \tag{8}$$

where $\hat{\mathbf{T}}_{kl}^s$ is computed based on the scene flow and segmentation, as described in the next paragraph.

**Segmentation & Flow $\rightarrow$ Motion $\hat{\mathbf{T}}_{kl}^s$.** Following previous works [32, 61], we employ the weighted-Kabsch algorithm [37, 27] to determine part-level rigid transformation $\hat{\mathbf{T}}_{kl}^s$ given the estimates of scene flow and rigid masks. Further, the motion head is optimized by the rotation component of $\hat{\mathbf{T}}_{kl}^s$, and estimates corresponding translation by minimizing our probability-based part-level distance. The implementation details of this distance and our motion loss can be found in **Supp**.

## 4 Experiments

### 4.1 Datasets & Metrics

Our model is evaluated on three various application scenarios with four datasets: 1) SAPIEN [68] for articulated objects, 2) OGC-DR and its single-view counterpart OGC-DRSV [61] for furniture arrangements; 3) KITTI-SF [48] for vehicular traffic. Following previous works [61, 32, 51], we evaluate the rigid segmentation performance on the following seven measures: Average Precision (**AP**), Panoptic Quality (**PQ**), F1-score (**F1**), Precision (**Pre**), and Recall (**Rec**) at an Intersection over Union threshold of 0.5, in addition to the mean Intersection over Union (**mIoU**) and Rand Index (**RI**). In terms of multi-body motion estimation, we report End-Point-Error 3D (**EPE3D**) in the main body, and other metrics (*e.g.*, Outlier) are provided in **Supp**. More information about the datasets and our evaluation protocol can be found in **Supp**.

### 4.2 Pilot Studies on The Two Heads

We first conduct experiments on SAPIEN to verify the generalizability of our two-head structure, including the point-level invariance of the segmentation head and the part-level equivariance of the motion head.

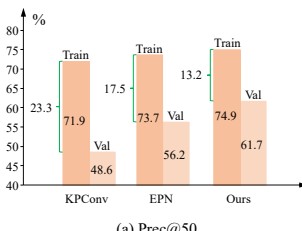 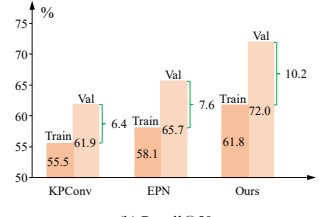 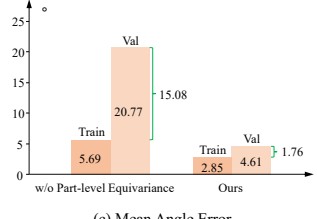

|  (a) Prec@50 | (b) Recall@50 | (c) Mean Angle Error |

Figure 4: *Results of pilot studies.* (a)(b) are the experimental results of our segmentation head, while (c) is the exploration outcomes of the motion head.

**Can the point-level invariance of our segmentation head help generalize to open-set motion?**
On SAPIEN, each articulated target has four frames with various part-level rigid orientations and locations. By training the segmentation head on a subset of the $1^{st}$ frames (with the canonical pose in [32]) and testing them on all frames, we compare its performance with the segmentation results using the same parameters of the non-equivariant counterpart KPConv [62] and global equivariant EPN. As shown in Figure 4(a)(b), the SE(3)-equivariance can essentially boost the generalization even given the global since there are some targets that only contain two rigid parts. By introducing point-level invariance, both precision and recall can be further improved.

**Can the part-level equivariance of our motion head help generalize to category-agnostic parts?**
Note that the category of targets for training, validation, and testing on SAPIEN is completely disjoint. By optimizing the motion head on the training set and testing its performance on the validation data, we ensure that the targets do not overlap in categories so their parts are entirely category-agnostic. To exclude the interference of part segmentation, we directly provide the ground-truth rigid partition as the mask. By comparing the decrease in performance on the validation data when using part-level equivariance *vs.* not using it, as illustrated in Figure 4(c), it is observed that the mean angular error of our model only increases by less than $2°$, while that without part-level equivariance surges from $5.69°$ to $20.77°$. This demonstrates the strong generalizability of our model to category-agnostic parts.

Table 1: *Ablation studies on SAPIEN.*

| Seg. Head | | Scene Flow | | Mot. Head | Metrics | | | | | | |
|---|---|---|---|---|---|---|---|---|---|---|---|
| SE(3) Feat. | Point-level Flexibility | Current | Past | Consensus | AP↑ | PQ↑ | F1↑ | Pre↑ | Rec↑ | mIoU↑ | RI↑ |
| | | | | | 45.2 | 44.2 | 58.9 | 53.8 | 65.1 | 60.9 | 71.2 |
| ✓ | | | | | 51.7 | 50.0 | 65.8 | 64.7 | 67.0 | 61.6 | 72.3 |
| ✓ | ✓ | | | | 55.3 | 52.8 | 68.3 | 65.9 | 70.0 | 62.3 | 72.7 |
| ✓ | ✓ | ✓ | | | 54.8 | 52.0 | 67.6 | 66.0 | 69.3 | 63.5 | 73.8 |
| ✓ | ✓ | ✓ | ✓ | | 57.0 | 51.6 | 67.3 | 63.8 | 71.1 | 63.1 | 73.2 |
| ✓ | ✓ | ✓ | ✓ | ✓ | 63.8 | 61.3 | 77.3 | 84.2 | 71.3 | 63.7 | 75.4 |

## 4.3 Ablation Studies

We perform detailed ablation studies on SAPIEN to explore the effects of three main components: segmentation head, motion head, and scheme of scene flow updating.

**Segmentation Head.** Two crucial design elements significantly enhance our segmentation head: 1) the use of SE(3)-equivariant features, and 2) point-level flexibility for invariant representations. By simply introducing the global equivariant features, AP increases from $45.2\%$ to $51.7\%$. This demonstrates the importance of SE(3)-equivariance in modeling rigid transformations. The proposed point-level flexibility further boosts the performance to $55.3\%$, suggesting that our point-level invariance is highly advantageous to the multi-body task.

**Scene Flow Updating Scheme.** For the updated choice of scene flow, we empirically test three options: 1) completely relying on the past flow extracted by an unsupervised flow network, 2) ignoring the past flow and directly using the current estimation, and 3) combining them together with a decay

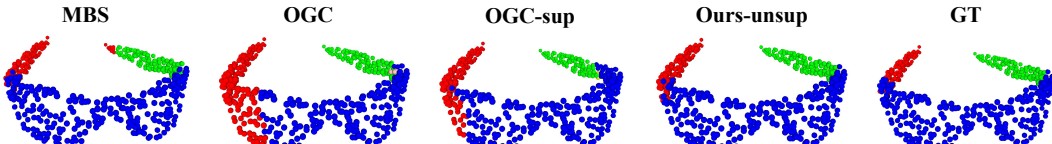

| MBS | OGC | OGC-sup | Ours-unsup | GT |

Figure 5: *Qualitative comparison between our **unsupervised** results and other methods (including supervised ones) on SAPIEN.* This provides a glimpse into the qualitative performance of our approach, and more results can be found in **Supp**.

factor. It is observed that the extreme choices of 1) or 2) result in minor performance differences (AP: 55.3% *vs.* 54.8%). The significant improvement comes from the weighted combination in option 3), achieving the AP of 57.0%.

**Motion Head.**   The motion head contributes to the consensus score for the removal of scene flow noise. From the last row of Table 1, the consensus between the motion head and scene flow significantly boosts the final segmentation performance in most of the metrics. By controlling the point-level learning weight during the training process, our motion head can assist in optimizing the segmentation head.

Table 2: *Rigid segmentation and motion estimation results on SAPIEN.* * indicates that we evaluate these metrics upon the officially released model; - means that the metric is unavailable.

|  |  | AP↑ | PQ↑ | F1↑ | Pre↑ | Rec↑ | mIoU↑ | RI↑ | EPE3D↓ |
|---|---|---|---|---|---|---|---|---|---|
| Supervised Methods | PointNet++ [55] | - | - | - | - | - | 51.2 | 65.0 | - |
|  | MeteorNet [46] | - | - | - | - | - | 45.7 | 60.0 | - |
|  | DeepPart [71] | - | - | - | - | - | 53.0 | 67.0 | 5.95 |
|  | MBS [32] | 49.4* | 52.6* | 67.6* | 61.4* | 75.2* | 67.3 | 77.0 | 5.03 |
|  | OGC-sup [61] | 66.1 | 48.7 | 62.0 | 54.6 | 71.7 | 66.8 | 77.1 | - |
|  | Ours-sup | **73.5** | **57.8** | **71.1** | **65.6** | **77.7** | **72.6** | **81.4** | **3.86** |
| Unsupervised Methods | TrajAffn [52] | 6.2 | 14.7 | 22.0 | 16.3 | 34.0 | 45.7 | 60.1 | - |
|  | SSC [51] | 9.5 | 20.4 | 28.2 | 20.9 | 43.5 | 50.6 | 65.9 | - |
|  | WardLinkage [66] | 17.4 | 26.8 | 40.1 | 36.9 | 43.9 | 49.4 | 62.2 | - |
|  | DBSCAN [19] | 6.3 | 13.4 | 20.4 | 13.9 | 37.9 | 34.2 | 51.4 | - |
|  | NPP [28] | - | - | - | - | - | 51.5 | 66.0 | 21.22 |
|  | OGC [61] | 55.6 | 50.6 | 65.1 | 65.0 | 65.2 | 60.9 | 73.4 | - |
|  | Ours | **63.8** | **61.3** | **77.3** | **84.2** | **71.3** | **63.7** | **75.4** | **5.47** |

## 4.4   Results & Comparisons

### 4.4.1   SAPIEN

SAPIEN is a challenging dataset of articulated objects since its training, validation and testing sets comprise completely disjoint categories of objects. This domain gap in the data split complicates the precise segmentation of rigid parts. Adapting our model to the supervised formulation is straightforward by optimizing the segmentation with ground-truth training labels. Therefore, we also report the supervised performance of the proposed framework. As shown in Table 2, our segmentation performance *sets a new benchmark across all metrics* in both supervised and unsupervised settings.

In line with the default setting of [32], we estimate motion between both consecutive frames and non-adjacent ones. Table 2 reports its EPE3D performance and comparison: our unsupervised performance of 5.47 EPE3D on motion estimation is on par with the best existing supervised method (MBS), while our supervised EPE3D achieves 3.86, marking a relative error reduction of approximately 23%.

As depicted in Figure 1, owing to the lightweight two-head structure, our model encompasses only a small number of parameters (0.25M) and incurs a low cost in computational complexity (0.92G floating point operations, FLOPs). We also show a sneak peek of the qualitative results (Figure 5). Our unsupervised method is comparable (and often times better) than other supervised methods. More results are shared in the **Supp**.

Table 3: *Rigid segmentation results on OGC-DR and OGC-DRSV.*

| | | AP↑ | PQ↑ | F1↑ | Pre↑ | Rec↑ | mIoU↑ | RI↑ |
|---|---|---|---|---|---|---|---|---|
| Supervised | OGC-sup [61] | 90.7 / 86.3 | 82.6 / 78.8 | 87.6 / 85.0 | 83.7 / 82.2 | 92.0 / 88.0 | 89.2 / 83.9 | 97.7 / 97.1 |
| Methods | Ours-sup | **92.8 / 89.3** | **86.9 / 82.6** | **91.0 / 87.9** | **88.8 / 85.5** | **93.2 / 90.4** | **91.2 / 86.6** | **98.7 / 97.9** |
| | TrajAffn [52] | 42.6 / 39.3 | 46.7 / 43.8 | 57.8 / 54.8 | 69.6 / 63.0 | 49.4 / 48.4 | 46.8 / 45.9 | 80.1 / 77.7 |
| | SSC [51] | 74.5 / 70.3 | 79.2 / 75.4 | 84.2 / 81.5 | 92.5 / 89.6 | 77.3 / 74.7 | 74.6 / 70.8 | 91.5 / 91.3 |
| Unsupervised | WardLinkage [66] | 72.3 / 69.8 | 74.0 / 71.6 | 82.5 / 80.5 | **93.9 / 91.8** | 73.6 / 71.7 | 69.9 / 67.2 | 94.3 / 93.3 |
| Methods | DBSCAN [19] | 73.9 / 71.9 | 76.0 / 76.3 | 81.6 / 81.8 | 85.8 / 79.1 | 77.8 / 84.8 | 74.7 / 80.1 | 91.5 / 93.5 |
| | OGC [61] | 92.3 / 86.8 | 85.1 / 77.0 | 89.4 / 83.9 | 85.6 / 77.7 | 93.6 / 91.2 | 90.8 / 84.8 | 97.8 / 95.4 |
| | Ours | **93.9 / 88.1** | **87.0 / 80.0** | **91.1 / 86.1** | 87.0 / 80.8 | **95.6 / 92.2** | **92.4 / 86.7** | **98.1 / 96.6** |

### 4.4.2 OGC-DR & Single-view Counterpart

OGC-DR is a dataset for indoor furniture arrangements, where the training, validation, and testing instances are distinct from one another. OGC-DRSV, the single-view version of OGC-DR, presents a challenge due to the incomplete nature of the furniture caused by occlusion, making it difficult to identify consistent rigidity. As demonstrated in Table 3, the proposed framework constantly surpasses the state-of-the-art models across all measures. Remarkably, even under the most challenging condition, *i.e.*, unsupervised single-view, our AP still achieves the value of 88.1%. This underscores the robustness of our model in handling incomplete observations. The performance on motion estimation can be found in **Supp**.

Table 4: *Rigid segmentation results on KITTI-SF.* Our model still achieves competitive results even though the data setting is inconsistent with the model's assumption.

| Method Category | Method | AP↑ | PQ↑ | F1↑ | Pre↑ | Rec↑ | mIoU↑ | RI↑ |
|---|---|---|---|---|---|---|---|---|
| Supervised | OGC-sup [61] | 62.4 | 52.7 | 65.1 | 63.4 | 67.0 | 67.3 | 95.0 |
| Methods | Ours-sup | **65.1** | **56.3** | **68.6** | **69.4** | **67.8** | **69.5** | **95.7** |
| | TrajAffn [52] | 24.0 | 30.2 | 43.2 | 37.6 | 50.8 | 48.1 | 58.5 |
| | SSC [51] | 12.5 | 20.4 | 28.4 | 22.8 | 37.6 | 41.5 | 48.9 |
| Unsupervised | WardLinkage [66] | 25.0 | 16.3 | 22.9 | 13.7 | **69.8** | 60.5 | 44.9 |
| Methods | DBSCAN [19] | 13.4 | 22.8 | 32.6 | 26.7 | 42.0 | 42.6 | 55.3 |
| | OGC [61] | **54.4** | 42.4 | 52.4 | 47.3 | 58.8 | **63.7** | **93.6** |
| | Ours | 53.6 | **44.4** | **55.1** | **56.3** | 54.0 | 61.5 | 93.4 |

### 4.4.3 KITTI-SF

Strictly speaking, KITTI-SF is not a multi-body rigid dataset, as the background in its point clouds may be deformable. This is inconsistent with the assumption of our framework, which posits that the feature is only equivariant for rigid transformations. However, as shown in Table 4, our framework still delivers competitive performance in rigid segmentation despite this inconsistency.

## 5  Conclusion

This paper introduces a part-level SE(3)-equivariant framework for modeling multi-body rigid motion. The two heads for rigid segmentation and motion estimation are meticulously designed based on their inherent invariant and equivariant characteristics. The relationship among scene flow, rigid segmentation, and multi-body transformation is then exploited to derive an unsupervised optimization strategy. Our approach achieves state-of-the-art results on multiple datasets while significantly reducing the required parameters and computations.

**Limitations & Future Work.**   Our model is predicated on an assumption: the part-level motion should be a rigid transformation. Therefore, as shown in Table 3, if the observation of part-level movements is non-rigid (such as occluded single views), it would suffer a performance decrease. In the future, we would like to develop a model that is more robust to its observation.

**Acknowledgements.** Our research is supported by Amazon Web Services in the Oxford-Singapore Human-Machine Collaboration Programme and by the ACE-OPS project (EP/S030832/1). We are grateful to all of the anonymous reviewers for their valuable comments.

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
