# Multi-body SE(3) Equivariance for Unsupervised Rigid Segmentation and Motion Estimation (Supplementary Material)

**Jia-Xing Zhong, Ta-Ying Cheng, Yuhang He, Kai Lu, Kaichen Zhou✉,**
**Andrew Markham, Niki Trigoni**
Department of Computer Science, University of Oxford
{jiaxing.zhong, ta-ying.cheng, yuhang.he, kai.lu, rui.zhou}@cs.ox.ac.uk
{andrew.markham, niki.trigoni}@cs.ox.ac.uk

## 1 Implementation Details of Network Structure

### 1.1 Per-point Feature Extractor

For an input frame $P_k$, the feature extractor of EPN outputs per-point SE(3)-equivariant representations $F_k \in \mathbb{R}^{N \times |\mathcal{G}| \times D}$. The corresponding feature $f_k^i \in \mathbb{R}^{|\mathcal{G}| \times D}$ of a point $p_k^i$ can be viewed as a concatenation of different representations w.r.t. $g_j \in \mathcal{G}$ over the rotation group dimension:

$$f_k^i = [\theta(p_k^i, g_1), \theta(p_k^i, g_2), \cdots, \theta(p_k^i, g_{|\mathcal{G}|-1}), \theta(p_k^i, g_{|\mathcal{G}|})], \tag{1}$$

where $\theta$ is a $D$-dimensional encoder based on a stack of convolution kernels. The prediction module of vanilla EPN is designed for global SE(3)-equivariance for all input points. Differently, our unsupervised multi-body task requires the model's ability to handle *part-level local equivariance*, especially under low-quality training signals. For this purpose, we further devise two heads for rigid segmentation and motion estimation. Figure 1 demonstrates the detailed design of our EPN feature extractor on SAPIEN, OGC-DR, and OGC-DRSV, and that on KITTI-SF shares the same structure but has larger numbers of output dimensions accordingly.

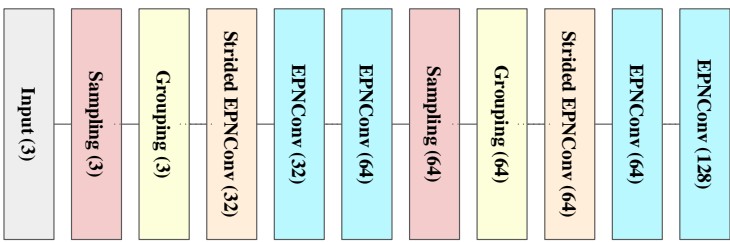

Figure 1: *Structure of our feature extractor based on EPN*. The number in brackets denotes the output dimension of the corresponding convolution/operation. "EPNConv" is the SE(3)-equivariant convolution proposed in the vanilla EPN network.

### 1.2 Point-level Invariant Segmentation Head

Rigid segmentation is an SE(3)-invariant task, as the predicted mask should remain consistent for the same points across various poses and positions. Traditional SE(3)-equivariant structures assume that all input points undergo the same rigid transformation, which is not in alignment with the multi-body setting. To encode distinct transformations for individual input points, the equivariant

37th Conference on Neural Information Processing Systems (NeurIPS 2023).

features $\theta(p_k^i, g_j)$ are aggregated into an invariant representation $u_k^i$ across the dimension of the rotation group:

$$u_k^i = \sum_j^{|\mathcal{G}|} w(p_k^i, g_j)\theta(p_k^i, g_j), \tag{2}$$

where $w(p_k^i, g_j) \in [0, 1]^{|\mathcal{G}|}$ is a selection probability of discrete rotations in $\mathcal{G}$, derived through a $1 \times 1$ convolution. The weighted sum $u_k^i$ is invariant to rigid motion given a point with its neighbors in the convolution receptive field. By fusing such invariant representations $u_{k(1)}^i \in \mathbb{R}^D, \cdots, u_{k(h)}^i \in \mathbb{R}^{D'}$ from $h$ layers in the feature extractor, the segmentation head outputs a soft prediction $\hat{M}_k \in [0, 1]^{N \times S}$ of the rigid mask. The multi-layer invariant representations are aggregated through an hourglass decoder, as shown in Figure 2.

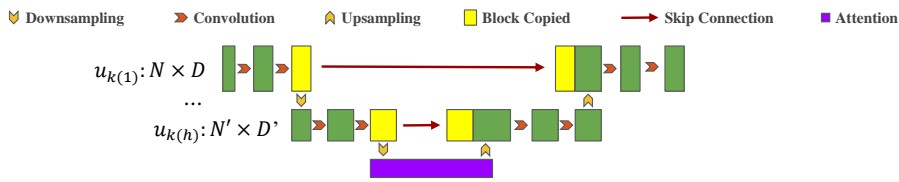

Figure 2: *Structure of multi-layer aggregation.* $N'$ and $D'$ are the point number and the feature dimension of the last invariant layer, respectively.

## 1.3 Part-level Equivariant Motion Estimation Head

Part-level SE(3)-equivariance is desirable for motion analysis, especially rotation estimation. Based on the noisy predictions $(\hat{M}_k, \hat{M}_l)$ of the frames $(k, l)$ from the head of rigid segmentation, the motion head is supposed to *handle these uncertain category-agnostic parts*. First of all, the part-level SE(3) feature $V_{k,j} \in \mathbb{R}^{N \times D \times S}$ w.r.t. the rotation $g_j \in \mathcal{G}$ of a single frame $P_k$ is obtained from the per-point equivariant representations $F_k$ and the predicted rigid mask $\hat{M}_k$:

$$V_{k,j} = \{\hat{m}_k^1 \cdot \theta(p_k^1, g_j), \hat{m}_k^2 \cdot \theta(p_k^2, g_j), \cdots, \hat{m}_k^{N-1} \cdot \theta(p_k^{N-1}, g_j), \hat{m}_k^N \cdot \theta(p_k^N, g_j)\}, \tag{3}$$

where $\hat{m}_k^i$ is the element corresponding to a point $p_i^k$ in $\hat{M}_k$, and $\cdot$ is the broadcast operation of Hardamard product. Afterward, $V_{k,j}$ over the rotation group $\mathcal{G}$ is concatenated as $V_k \in \mathbb{R}^{N \times |\mathcal{G}| \times D \times S}$, followed by a permutation-invariant operation $\sigma : \mathbb{R}^{N \times |\mathcal{G}| \times D \times S} \to \mathbb{R}^{|\mathcal{G}| \times D \times S}$ (*e.g.*, max pooling) over all the points to produce part-level equivariant features $\tilde{V}_k = \sigma(V_k)$. Between two frames $(k, l)$, the part-level rotation correlated feature $C_{kl} \in \mathbb{R}^{|\mathcal{G}| \times |\mathcal{G}| \times S}$ is defined as:

$$C_{kl} = \tilde{V}_k \tilde{V}_l^T, \tag{4}$$

where $^T$ is matrix transposition. $C_{kl}$ is calculated upon "softly matching" within each consistent rigid part, while the specific category labels can be agnostic to the model. Based on the correlated feature $C_{kl}$, the motion head estimates rotation $\hat{\mathbf{R}}_{kl}^s$ and translation $\hat{\mathbf{t}}_{kl}^s$ of each rigid part $s$. To be specific, the rotation regression is implemented through a $1 \times 1$ convolution. Unlike the confidence-based selection in vanilla EPN for single-object pose regression, we choose the anchor $g_{kl}^s$ from $\mathcal{G}$ by minimizing the registration error and then optimize the residual $r_{kl}^s$. In this case, the rotation is computed as:

$$\hat{\mathbf{R}}_{kl}^s = g_{kl}^s r_{kl}^s. \tag{5}$$

The translation can be derived from the minimal weighted distance between the transformed frame of $P_k$ and the origin $P_l$:

$$\hat{\mathbf{t}}_{kl}^s = \underset{\mathbf{t}}{\arg\min}\, d(\hat{\mathbf{R}}_{kl}^s P_k + \mathbf{t}, P_l), \tag{6}$$

where $d$ is the chamfer loss weighted by the mask predictions.

# 2 Implementation Details of Unsupervised Training Strategy

## 2.1 Hyper-settings of Model Training

The training settings for the model include a total of $40$ epochs and a batch size of $16$. The learning rate is initially set to $2.0 \times 10^{-4}$ and decays at a rate of $0.7$ with a minimum clip value of $1.0 \times 10^{-4}$. The batch normalization momentum is set to $0.9$, which controls the smoothing of the batch normalization statistics. The decay step is set to $2.0 \times 10^5$, which determines the frequency at which the learning rate decays.

## 2.2 Segmentation & Flow $\rightarrow$ Motion $\hat{\mathbf{T}}_{kl}^s$

Following previous works [6, 10], we employ the weighted-Kabsch algorithm [7, 5] to determine part-level rigid transformation $\hat{\mathbf{T}}_{kl}^s$ given the estimates of scene flow and rigid masks. Further, the motion head is optimized by the rotation component of $\hat{\mathbf{T}}_{kl}^s$, and estimates corresponding translation by minimizing our probability-based part-level distance as follows:

$$d_{prob}^s = d(\hat{\mathbf{T}}_{kl}^s \circ (\hat{M}_k^s P_k), \hat{M}_l^s P_l), \tag{7}$$

where $d$ is the chamfer loss. $\hat{M}_k^s$ and $\hat{M}_l^s$ is the predicted rigid masks of the $s^{th}$ part for the frame $P_k$ and $P_l$, respectively.

# 3 Datasets & Metrics

## 3.1 Datasets

**SAPIEN**, as described by [11], provides a collection of $720$ simulated articulated objects with annotations at the part instance level. Each object is represented by $4$ sequential scans, with part instances exhibiting varying articulating states. Following the methodology of Huang *et al.* [6], we utilize the training data generated by Yi *et al.* [12]. Specifically, the dataset consists of $82092$ pairs of point clouds for training and $2880$ single point cloud frames for testing. Both training and testing point clouds are downsampled to $512$ points.

**OGC-DR**, proposed by Song and Yang [10], is applicable to both scene flow estimation and object segmentation tasks. By adhering to the approach outlined by [9], they randomly positioned $4$ to $8$ objects from $7$ classes of the ShapeNet dataset [2], including chairs, tables, lamps, sofas, cabinets, benches, and displays, within each room. A total of $3750$ indoor rooms were generated for training purposes, with an additional $250$ for validation and $1000$ for testing. Rigid dynamics were introduced within each scene by applying continuous random transformations to each object and capturing $4$ sequential frames for evaluation purposes. Each point cloud frame was subsequently downsampled to $2048$ points. Song and Yang utilized the methodology proposed by Choy *et al.* [3] to partition different object instances across train/val/test sets.

**OGC-DRSV**, expanded upon the OGC-DR dataset by Song and Yang [10], is collected from single depth scans at each time step on the mesh models, designated as Single-View OGC-DR. Due to self- and/or mutual occlusions, all object point clouds within OGC-DRSV are severely incomplete, rendering this new dataset considerably more challenging than its predecessor, OGC-DR. Each point cloud frame within OGC-DRSV was also downsampled to $2048$ points.

**KITTI-SF** comprises $200$ pairs of point clouds from real-world traffic scenes for training purposes [8], as well as an online hidden test for scene flow estimation. Following [10], we trained our pipeline on the first $100$ pairs of point clouds and subsequently tested it on the remaining $100$ pairs ($200$ single point clouds). During the testing phase, only the human annotations of cars and trucks within each frame are retained for score computation. All other objects were considered part of the background. The entire background was not disregarded, but rather treated as a single object in our evaluation. Additionally, cars and trucks could be either static or dynamic.

## 3.2 Metrics

**Rigid Segmentation.** Average Precision (**AP**) is a measure that takes into account both precision and recall over all labels. It is calculated as the mean of the precision values at different recall levels.

Panoptic Quality (**PQ**) is a measure proposed for evaluating panoptic segmentation, which takes into account both recognition and segmentation quality. F1-score (**F1**) is the harmonic mean of precision and recall. Precision (**Pre**) measures the proportion of true positive instances among the instances that were predicted as positive by the model, while Recall (**Rec**) measures the proportion of true positive instances that were correctly identified by the model. Mean Intersection over Union (**mIoU**) is a measure used to evaluate semantic segmentation models, which calculates the average IoU between the predicted and ground truth segmentation masks for each class. Rand Index (**RI**) is a measure of similarity between two data clusterings, which can be used to evaluate the performance of a segmentation model by comparing its predictions with the ground truth masks.

**Motion Estimation.** End-Point-Error 3D (**EPE3D**) measures the average error over all warped points under the estimated and the ground-truth warping functions. Accuracy Strict (**AccS**) and Accuracy Relaxed (**AccR**) refer to the ratio of points where the EPE3D or relative error is below a certain threshold. Specifically, AccS represents the ratio of points where the EPE3D is less than 0.05 or the relative error is less than 0.05, while AccR represents the ratio of points where the EPE3D is less than 0.1, or the relative error is less than 0.1. Outlier (**Outl**) refers to the ratio of points where the EPE3D is greater than 0.3 or the relative error is greater than 0.1.

# 4 Rigid Motion Estimation Performance: More Metrics & More Datasets

Figure 3 shows the results of the motion estimation experiments on SAPIEN. The fully-supervised method outperforms the unsupervised method in terms of all of the metrics. The fully-supervised method achieved an EPE3D of 3.86, an AccS of $42.75\%$, an AccR of $63.58\%$, and an Outlier rate of $56.12\%$ while the unsupervised method achieved an EPE3D of 5.47, an AccS of $32.76\%$, an AccR of $52.81\%$ and an Outlier rate of $66.59\%$.

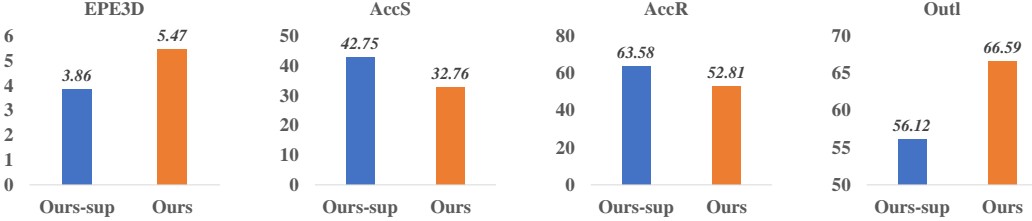

Figure 3: *Motion estimation results on SAPIEN.*

Figure 4 demonstrates the results of the motion estimation experiments on OCG-DR. Intriguingly, the unsupervised performance is marginally better than the fully-supervised case on this dataset, possibly because full supervision constrains the generalizability of the model of motion estimation. The fully-supervised method achieved an EPE3D of 0.91, an AccS of $67.68\%$, an AccR of $89.11\%$ and an Outlier rate of 57.38, while the unsupervised method achieved an EPE3D of 0.73, an AccS of $76.38\%$, an AccR of $92.96\%$ and an Outlier rate of $45.20\%$.

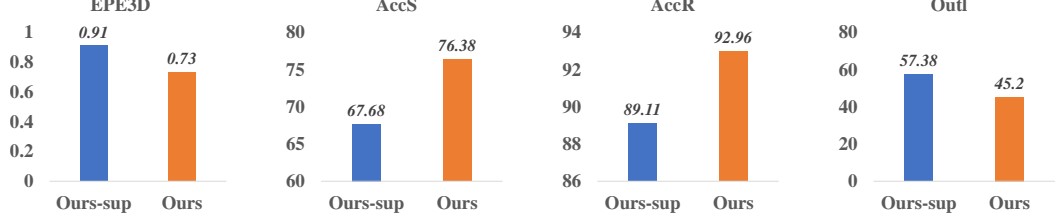

Figure 4: *Motion estimation results on OGC-DR.*

As depicted in Figure 5, the motion estimation experiments conducted on OCG-DRSV demonstrate that the fully-supervised method surpasses the unsupervised method. The fully-supervised method attained an EPE3D of 0.87, an AccS of $75.49\%$, an AccR of $92.91\%$, and an Outlier rate of $59.43\%$. In contrast, the unsupervised method attained an EPE3D of 1.45, an AccS of $61.27\%$, an AccR of $81.82\%$, and an Outlier rate of $65.39\%$.

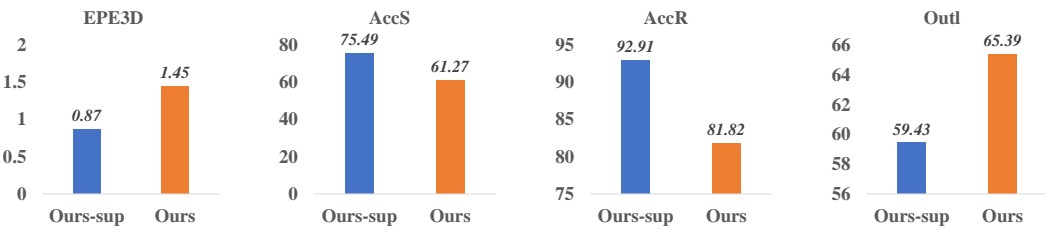

Figure 5: *Motion estimation results on OGC-DRSV.*

# 5 More Quantitative Results on KITTI-Det and SemanticKITTI

Following the same evaluation protocol of OGC, we further evaluate the generalizability of our model from KITTI-SF to two larger outdoor datasets, i.e., KITTI-Det [4] and SemanticKITTI [1], and compare the performance with the results reported in OGC, as shown in Table 1& 2.

Table 1: *Segmentation performance on KITTI-Det.*

| Methods | AP↑ | PQ↑ | F1↑ | Pre↑ | Rec↑ | mIoU↑ | RI↑ |
|---|---|---|---|---|---|---|---|
| OGC-sup [10] | 51.4 | 41.0 | 49.1 | 43.7 | 56.0 | 66.2 | 91.0 |
| Ours-sup | **52.5** | **43.3** | **51.8** | **47.5** | **57.0** | **68.0** | **92.6** |
| OGC-unsup [10] | 40.5 | 30.9 | 37.0 | 30.8 | **46.5** | **60.6** | 86.4 |
| Ours-unsup | **41.3** | **32.9** | **38.8** | **35.3** | 43.1 | 60.2 | **87.2** |

Table 2: *Segmentation performance on SemanticKITTI.*

| Sequences | Methods | AP↑ | PQ↑ | F1↑ | Pre↑ | Rec↑ | mIoU↑ | RI↑ |
|---|---|---|---|---|---|---|---|---|
| 00 - 10 | OGC-sup [10] | 53.8 | 41.3 | 48.1 | 40.1 | 60.0 | 68.3 | 90.0 |
| | Ours-sup | **60.1** | **47.6** | **55.4** | **48.6** | **64.4** | **71.9** | **93.4** |
| | OGC-unsup [10] | 42.6 | 30.2 | 35.3 | 28.2 | 47.3 | 60.3 | 86.0 |
| | Ours-unsup | **46.9** | **31.6** | **36.9** | **29.0** | **50.6** | **63.2** | **88.7** |
| 00 - 07 & 09 - 10 | OGC-sup [10] | 55.3 | 41.8 | 48.4 | 40.1 | 61.1 | 69.9 | 90.3 |
| | Ours-sup | **60.5** | **48.1** | **55.6** | **48.8** | **64.7** | **73.2** | **93.8** |
| | OGC-unsup [10] | 43.6 | 30.5 | 35.5 | 28.1 | 48.2 | 62.1 | 86.3 |
| | Ours-unsup | **47.4** | **31.7** | **36.8** | **28.7** | **51.0** | **64.8** | **89.3** |
| 08 | OGC-sup [10] | 49.4 | 39.2 | 46.6 | 40.0 | 55.8 | 60.3 | 88.3 |
| | Ours-sup | **58.4** | **46.0** | **54.4** | **47.8** | **63.1** | **65.8** | **91.7** |
| | OGC-unsup [10] | 38.6 | 29.1 | 34.7 | 28.6 | 44.0 | 51.8 | 84.3 |
| | Ours-unsup | **44.2** | **31.0** | **37.3** | **30.0** | **49.1** | **55.8** | **86.1** |

# 6 Qualitative Results

Figure 6 presents the qualitative results of our model. In general, the predictions generated by the model are satisfactory. However, errors are primarily observed in the segmentation boundaries. For instance, the border between the tap handle and the faucet spout in the $2^{nd}$ picture is one such area where errors occur.

We also provide additional visualizations in Figure 7: 1) challenging scenes (multiple parts) on SAPIEN, with successful and failure cases (errors as circled in green on the final two rows); 2) visualizations on all of the other datasets (OGC-DR, OGC-DRSV, and KITTI-SF). All of the results ("Ours") are obtained from our model in the unsupervised setting, while "GT" means the ground-truth segmentation.

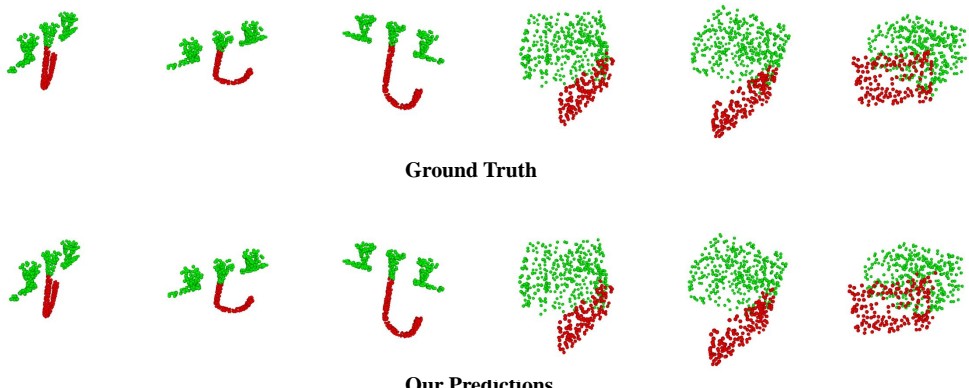

**Ground Truth**

**Our Predictions**

Figure 6: *Visualizations for rigid segmentation results on SAPIEN.*

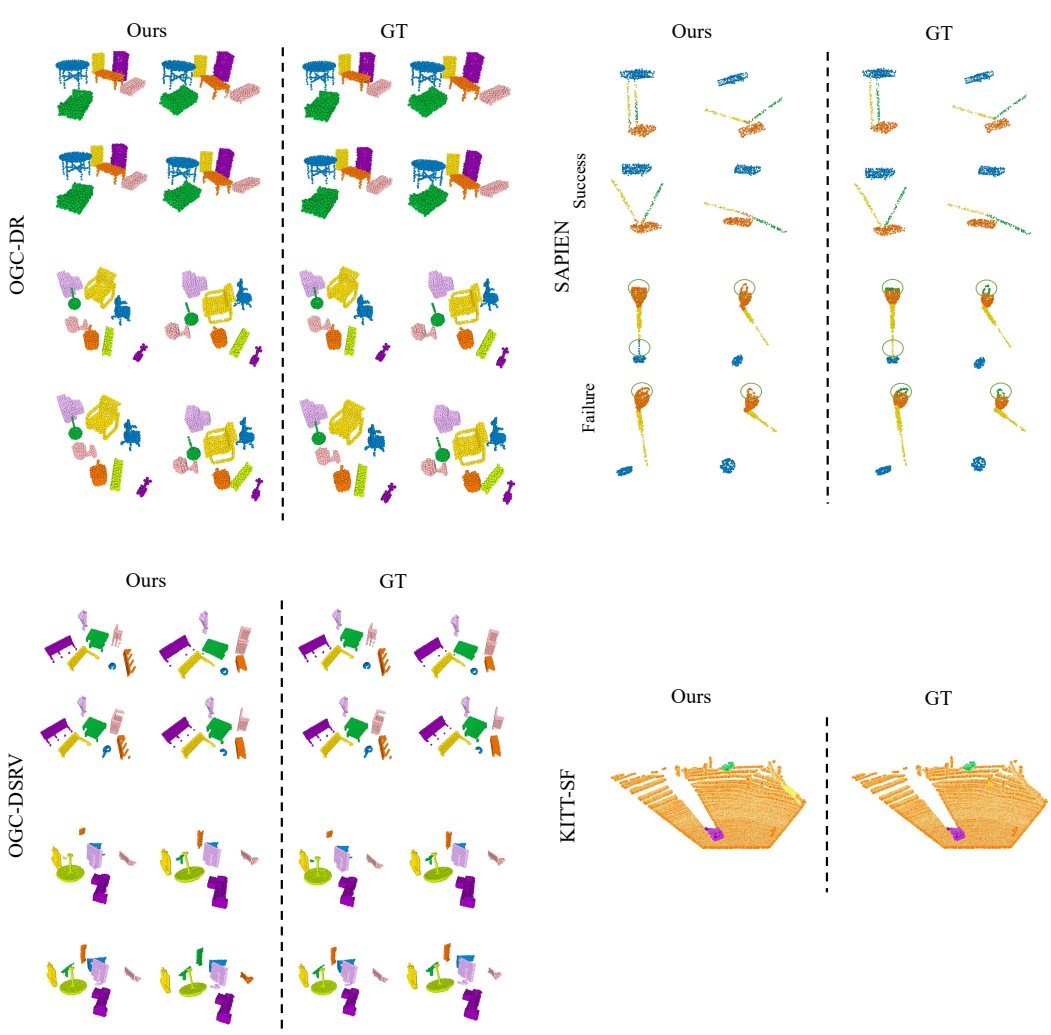

Figure 7: *Additional visualizations for challenging scenes and more datasets.*

# 7 Parameter Number and Computational Complexity

Table 3 provides a quantitative comparison of various supervised and unsupervised methods in terms of parameter number, FLOPs, and Average Precision on the SAPIEN dataset. The results demonstrate that the proposed supervised method (Ours-sup) achieves the highest Average Precision of 73.5 while utilizing the fewest parameters (0.25M) and FLOPs (0.92G). Similarly, the proposed unsupervised method (Ours) outperforms the OGC unsupervised method [7] with respect to Average Precision (63.8 *vs.* 55.6) while requiring fewer FLOPs (3.71G *vs.* 4.09G).

Table 3: *Quantitative results of parameter number, FLOPs, and Average Precision (AP) on SAPIEN. In this table, * indicates that we evaluate its AP upon the officially trained model.

|  |  | **FLOPs (G)**↑ | **#PARAMS (M)**↑ | **AP (%)**↓ |
|---|---|---|---|---|
| Supervised Methods | MBS [6] | 608.73 | 17.15 | 49.4* |
|  | OGC-sup [10] | 4.09 | 0.43 | 66.1 |
|  | Ours-sup | **0.92** | **0.25** | **73.5** |
| Unsupervised Methods | OGC [10] | 4.09 | 0.43 | 55.6 |
|  | Ours | **3.71** | **0.31** | **63.8** |