# OpenReview forum: "Multi-body SE(3) Equivariance for Unsupervised Rigid Segmentation and Motion Estimation"
_NeurIPS.cc/2023/Conference — NeurIPS 2023 poster_

### Official Review · Reviewer_XLXX · 2023-07-03

**Soundness:** 3 good
**Presentation:** 2 fair
**Contribution:** 3 good
**Rating:** 5
**Confidence:** 5

**Summary:**

The paper describes a method to simultaneously segment rigid parts and estimate part-level transformations for multi-body point cloud inputs. Given the input point cloud, equivariant features are first extracted and aggregated to invariant features for part segmentation. To estimate the transformations between two point clouds, an equivariant motion estimation head is applied to pairs of point clouds to obtain SE(3) motions. The entire model is trained in an unsupervised fashion using the consensus between 3D scene flow, motion segmentation, and transformations. Experiments show that the model is generalizable in two aspects: open-set motion and category-agnostic part segmentation. The authors test the model on SAPIEN, OGC, and KITTI datasets, and show superior performance in comparison to past baselines, with much better accuracy and much fewer FLOPs, even surpassing the supervised method.

**Strengths:**

1. The identification of two generalization problems, i.e. invariant segmentation and equivariant motion estimation, is insightful. The use of equivariant backbones and their invariant counterpart is well-suited for such a problem.
2. The use of initial scene flow to bootstrap (or cold-start) the learning process and to get rid of local minima is novel. By utilizing the consistency between the output attributes which is inherent in the multi-body registration task, the method could be learned in an unsupervised manner.
3. Experiments are extensive and the results are compelling. The method is tested on four different datasets ranging from articulated objects to outdoor driving scenes, justifying the wide applicability. In the meanwhile, the fact that the algorithm is able to achieve a much higher accuracy with a much lower cost is impressive and set up a strong waypoint for future works.

**Weaknesses:**

1. The unsupervised training strategy lacks clarity. What is the loss function? Is it eq.(8) plus eq.(7) in the supplementary with some balancing factor? This should be clarified in the main paper. In Line 213, the authors mention the weighted-Kabsch algorithm, what is the 'weight' here?
2. The generalization of the segmentation head is questionable -- this is my most important doubt about the paper. Although the generalization to category-agnostic part segmentations is demonstrated through experiments, the reason why an equivariant network backbone could achieve that is not explained. What adds to the confusion is that the method is trained in an unsupervised manner, which means that no canonical segmentation labels are defined. Given a novel multi-body shape that is not seen from the training set (e.g. a cabinet with four drawers), how could the model give a unique index to each of the drawers in a consistent fashion (i.e. the drawer index stays unchanged across multiple articulation states)? What is the maximum number of parts supported in the algorithm? Is it predefined or optimized/given for each test sample? The doubt also links to the experiment section -- In Fig.4 (a,b), the gap between training and validation is not shown, and it's unclear whether Fig.4 (c) is based on ground truth or predicted segmentation.
3. The experiments are not explained very clearly. In Sec4.2, why 'point-level invariance of segmentation' helps 'open-set motion' generalization, and why 'part-level equivariant of motion' helps 'category-agnostic part' generalization? Shouldn't they be swapped? It is preferable to move Sec4.4 ahead of Sec4.3.
4. More visualizations and challenging cases should be shown. There are only 3 visualizations provided in the paper+supplementary, on SAPIEN datasets. These visualizations only contain segmentation labels with two parts. To make the results more convincing, the author should provide more visualizations on all the datasets they've tested on, with transformations shown, and on more challenging cases, e.g., large scenes with >4 rigid parts.

**Questions:**

1. Does the current training scheme leverage the cycle consistency among multiple inputs? If not how could this be possibly incorporated into the algorithm?
2. How does the method generalize to different point densities/distributions? What is the maximum number of points could the method process?
3. Some typos:
- Line 144, 'There are mainly comprised of...'
- Line 237, 'even given the global since...'

**Limitations:**

Limitations are discussed in the last paragraph of the main paper.

---

> ### Author Rebuttal · Authors · 2023-08-10
>
> Thank you for your valuable feedback. We appreciate your recognition of our use of equivariant backbones & their invariant counterparts, our solution to the cold-start problem, and our extensive and compelling experiments. We would like to address each of your concerns in detail below.
>
> **Q1:** The unsupervised training strategy lacks clarity. What is the loss function? Is it eq.(8) plus eq.(7) in the supplementary with some balancing factor? This should be clarified in the main paper. In Line 213, the authors mention the weighted-Kabsch algorithm, what is the 'weight' here?
>
> **A1:** Thank you for bringing the clarity issues to our attention. We would like to address them as follows:
>
> (1) We apologize for your confusion caused by a missing term $\hat M_k^s$ in Equation 8, of which the correct unsupervised segmentation loss should be $l_{seg} = \frac{1}{NS} \sum^N_i \sum^S_s ||\beta_{kl}^s \hat{M}\_k^s ({p^\prime}^i_l  - \hat{\mathbf{T}}^s_{kl} \circ p^i_k)||$. By minimizing $l_{seg}$, the predictive part-level mask $\hat M_k^s$ can be optimized in a differentiable fashion.
>
> (2) We mainly utilize three tricks to balance the optimization process between the segmentation head (Eq. 8) and the motion head (Supp-Eq. 7): a) smoothly updating scene flow (Lines 194 - 200) mitigates the sharp fluctuation of motion estimation; b) the consensus $\beta^{s(i)}_{kl}$ (Lines 205 - 210) alleviates the influence of predictive errors from either head; c) Two heads are alternately updated per batch to prevent their gradients interfering each other, and their initial learning rates are set as segmentation : motion = 2.0e-4 : 2.0e-5.
>
> (3) The weight in the Kabsch algorithm refers to the weight assigned to the deviation of each point. Following [32, 61], we use the predictive mask of rigid segmentation as the point-wise weight.
>
> To address these concerns, we will take the following actions:
>
> (1) Rectifying Equation 8 and providing a description of the training strategy after Line 211.
>
> (2) Adding additional elaboration in Sections 2.1 and 2.2 within the supplementary material.
>
> (3) Introducing the pseudo-code for the weighted-Kabsch algorithm [37, 27] to the supplementary material and releasing our open-source implementation for public accessibility upon the publication of our work.
>
> **Q2:** Although the generalization to category-agnostic part segmentations is demonstrated through experiments, the reason why an equivariant network backbone could achieve that is not explained. What adds to the confusion is that the method is trained in an unsupervised manner, which means that no canonical segmentation labels are defined. Given a novel multi-body shape that is not seen from the training set (e.g. a cabinet with four drawers), how could the model give a unique index to each of the drawers in a consistent fashion (i.e. the drawer index stays unchanged across multiple articulation states)? What is the maximum number of parts supported in the algorithm? Is it predefined or optimized/given for each test sample? In Fig.4 (a, b), the gap between training and validation is not shown, and it's unclear whether Fig.4 (c) is based on ground truth or predicted segmentation.
>
> **A2:** (1) Segmentation Consistency: The segmentation head can take a single frame as its input, and the output part-level label indicates the partition inside a frame. The segmentation indexes are not necessarily consistent between the two frames. Following the common practice in [32, 61], we utilize the Hungarian-matching algorithm based on the IoU score between two masks to determine consistent parts. After the Hungarian-matching algorithm, the same rigid index across different frames is assigned together, and the correlated features $C_{kl}$ can be obtained for part-level equivariance in the motion head. We will include this description in Section 3.3 in the main body.
>
> (2) Maximum Number of Parts: This upper bound of part number is predefined as a sufficiently large number. Following the settings of OGC, the maximum is set as 8, 8, 8 and 15 on SAPIEN, OGC-DR, OGC-DRSV and KITTI-SF, respectively.
>
> (3) Explanation of Fig.4: a) The gap between training and validation is as follows:
>
> | Model  | Prec@50$\uparrow$ (Training) | Recall@50$\uparrow$ (Training) | Prec@50$\uparrow$ (Validation) | Recall@50$\uparrow$ (Validation) |
> |--------|------------------------------|--------------------------------|--------------------------------|----------------------------------|
> | KPConv | 71.9                         | 55.5                           | 48.6                           | 61.9                             |
> | EPN    | 73.7                         | 58.1                           | 56.2                           | 65.7                             |
> | Ours   | 74.9                         | 61.8                           | 61.7                           | 72.0                             |
>
>
> The performance gap between training and validation for our model is significantly smaller than that of other models, providing further evidence of its generalizability.
>
> b) For a fair comparison, the evaluation of our motion head excludes the inference from part segmentation, and both experiments in Fig.4 (c) are based on ground-truth segmentation.
>
> **Our response to your Q3-Q7 will be uploaded once the function of interactive comment is available.**

---

> > ### Author Response · Authors · 2023-08-10
> > **Response to Reviewer XLXX (Q3-Q7)**
> >
> > **Q3:** In Sec4.2, why 'point-level invariance of segmentation' helps 'open-set motion' generalization, and why 'part-level equivariant of motion' helps 'category-agnostic part' generalization? Shouldn't they be swapped? It is preferable to move Sec4.4 ahead of Sec4.3.
> >
> > **A3:** Thank you for helping us improve the paper writing. We will revise the paper to clarify the experiments and their explanations with the following three details:
> >
> > (1) Point-level invariance of our segmentation head makes the model robust to unseen pose changes (feature remains relatively consistent despite pose changes), which improves the generalizability to open-set motion (unknown pose variations). Part-level equivariance of the motion head helps generalize to category-agnostic parts because it allows the generalization based on the part-level motion, even if the category is unknown. Nevertheless, by improving the segmentation, the part-level equivariant motion head naturally obtains better supervision from the accurate mask, and vice versa.
> >
> > (2) Your suggestion of moving Sec4.4 ahead of Sec4.3 indeed creates a smoother logical flow from the intact model (Sec 4.4) to the ablations on separate modules (Sec 4.3). We will make this amendment to our paper accordingly.
> >
> > **Q4:** To make the results more convincing, the author should provide more visualizations on all the datasets they've tested on, with transformations shown, and on more challenging cases.
> >
> > **A4:** We provide additional visualizations (attached as PDF in the global response) across four datasets: OGC-DR, OGC-DSRV, SAPIEN, and KITTI-SF with multiple rigid parts. We also show some failure cases in challenging scenes on SAPIEN, as circled in green on the final two rows of the SAPIEN visualization. Please refer to the main response for more details on visualization.
> >
> > **Q5:** Does the current training scheme leverage the cycle consistency among multiple inputs? If not how could this be possibly incorporated into the algorithm?
> >
> > **A5:** Yes, we attempt to make use of cycle consistency from two aspects: (1) For a pair of frames $(P_k, P_l)$, scene flow is updated and obtained for both forward ($\hat\delta_{kl}$) and backward ($\hat\delta_{lk}$) directions. (2) The estimate of our motion head of the original input pair $(P_k, P_l)$ is rectified by averaging from the inversion of the predictive transformation from $(P_l, P_k)$. For readability， we simplify the description of cycle consistency; we will make sure to release the open-source implementation with these training details.
> >
> > **Q6:** How does the method generalize to different point densities/distributions? What is the maximum number of points could the method process?
> >
> > **A6:** (1) We follow standard protocols on corresponding datasets and evaluate our model under various point numbers. Specifically, the point number of one frame is set to 512, 2048, 2048 and 8192 on SAPIEN, OGC-DR, OGC-DRSV and KITTI-SF, respectively.
> >
> > (2) As for the generalizability to different point densities, we include additional experiments on KITTI-Det & SemanticKITTI, of which the points are sparser than those on KITTI-SF. Please refer to **A3** in our response to Reviewer ZttT for more details.
> >
> > **Q7:** Some typos.
> >
> > **A7:** Thank you for bringing these typos to our attention. We will make sure to correct the typos in Sec 3.3 & 4.2 in the revised manuscript.

---

> > > ### Comment · Reviewer_XLXX · 2023-08-15
> > >
> > > Dear authors, thank you for the detailed rebuttal. I've read all of them.
> > >
> > > I am still a bit worried about the generalizability of the method. If you train your model on SAPIEN, and test it on KITTI-SF, what will happen? Could you draw an intuitive boundary on the safe zone where the model would perform well?

---

> > > > ### Author Response · Authors · 2023-08-15
> > > > **Response to Reviewer XLXX**
> > > >
> > > > Dear Reviewer XLXX,
> > > >
> > > > Thank you very much for the rigor and effort you put into our rebuttal. In response to your question about the generalizability of our method, we would like to address it from two perspectives: 1) the intuitive boundary on how well the model could generalize, and 2) the reasonings behind why training on SAPIEN and testing on KITTI-SF isn't feasible.
> > > >
> > > > 1) Since our model can be trained in an unsupervised manner, it is agnostic to the specific parts that the training set comprises. Thus, your question regarding its generalizability and "intuitive boundary on the safe zone" is a very valuable insight that should be explored. To answer this, we will like to draw your attention to the composition of the SAPIEN dataset, Figure 4 in the main paper, and our rebuttal answer **A2**. The SAPIEN dataset is composited such that the training, validation, and testing subsets are all comprised of **different categories without overlap**:
> > > >
> > > > | SAPIEN Subset | Category                                                                                                                                                                                |
> > > > |---------------|-----------------------------------------------------------------------------------------------------------------------------------------------------------------------------------------|
> > > > | Training      | Table, Chair, Plane, Car, Guitar, Bike, Suitcase                                                                                                                                        |
> > > > | Validation    | Lamp, Pistol, Mug, Skateboard, Earphone, Rocket, Cap                                                                                                                                    |
> > > > | Testing       | Box, Dishwasher, Display, Storage Furniture, Eyeglasses, Faucet, Kettle, Knife, Laptop, Lighter, Microwave, Oven, Phone, Pliers, Safe, Stapler, Door, Toilet, Trash Can, Washing Machine |
> > > >
> > > > Given the fact that SAPIEN has entirely non-overlapping categories among its subsets, the experimental results in Figure 4 and our rebuttal answer **A2** that show minor discrepancies between training and validation errors, it would be safe to draw an intuitive boundary on the fact that our model would perform well in **various objects of similar sizes and even with unseen categories and rigid transformation**, much better than previous state-of-the-art methods like OGC.
> > > >
> > > > 2) This leads to our answer to the second question. We can imagine that the model trained on SAPIEN and tested on KITTI-SF would not generalize well. This is **not because of the lack of generalization**, but due to the significant difference in point cloud coordinate ranges between the two datasets. For example, a (normalized) ‘mug’ on SAPIEN has a much smaller size than a ‘truck’ on KITTI-SF. Consequently, it is improbable that our model, trained on such small mugs, would perform well on extremely large trucks. Quantitatively, we further analyze the maximum coordinate differences for the two datasets, the following results present the large gap in terms of the scene/object sizes of the two datasets:
> > > >
> > > > | Dataset  | $\Delta x_{max}$ | $\Delta y_{max}$ | $\Delta z_{max}$ |
> > > > |----------|------------------|------------------|------------------|
> > > > | SAPIEN   | 2.03             | 2.05             | 2.07             |
> > > > | KITTI-SF | 60.05            | 8.57             | 33.48            |
> > > >
> > > > For SAPIEN, the change in point cloud coordinates is 2.03-2.07 while for KITTI-SF it is 8.57-60.05: a vast difference of 4.18-29.58 times. This is also the main reason why no previous work has attempted to evaluate a model under this setting. Indeed, a work exploring how different point cloud ranges could be included in the same model would be an interesting direction forward, but it would be out of the scope and contribution of this work. We will, however, discuss this in our Future Work section.
> > > >
> > > > Therefore, we respectfully request you to give more credit to the SAPIEN training and validation results, as this comparison eliminates the problem of different point cloud sizes but at the same time shows the robustness of our model by performing consistently across its completely distinct training, validation, and testing categories. Thank you again for your thorough review and insightful comments.

---

> > > > > ### Comment · Reviewer_XLXX · 2023-08-20
> > > > >
> > > > > I'm sorry for the late reply due to my busy schedule. This is to confirm that I've taken your reply seriously and have reconsidered my rating.
> > > > >
> > > > > However, I'm still not fully convinced that the method will work out-of-box given solely the experimental results. Also in theory you could re-scale the KITTI dataset to make $\Delta_{x,y,z}$ matches. Importantly, the multi-body scheme is motion-based, rather than geometry-based, and it is impossible to infer the segmentation mask given a single frame without considering its motion.
> > > > >
> > > > > Nevertheless, I wouldn't consider decreasing my rating given the detailed response and I don't want to make a too aggressive judgment over this work whose presentation is quite nice indeed. I would rather keep my original rating.
> > > > >
> > > > > Thanks!

---

> > > > > > ### Author Response · Authors · 2023-08-21
> > > > > > **Response to Reviewer XLXX**
> > > > > >
> > > > > > Thank you for your reply and your honest feedback. We appreciate your diligence and professionalism in reviewing our paper. We are glad that you liked the presentation of our paper and that you decided to keep your positive rating.
> > > > > >
> > > > > > We would also like to clarify some points that might help you better understand our work. Our segmentation head can infer the rigid partition given only one frame as the input without any post-processing operations, but the model needs multiple frames to obtain consistent rigid IDs in the post-processing steps for motion estimation.
> > > > > >
> > > > > > We do not intend to argue with you or change your mind but hope that these clarifications can address some of your doubts. We sincerely thank you again for your valuable comments and suggestions.

---

### Official Review · Reviewer_S5HY · 2023-07-06

**Soundness:** 3 good
**Presentation:** 3 good
**Contribution:** 3 good
**Rating:** 6
**Confidence:** 4

**Summary:**

This paper studies the problem of rigid part segmentation and motion segmentation. The proposed network first uses a global SE(3) equivariant backbone to extract point features, then a part segmentation head will predict invariant zero-order segmentation and a motion head will predict one-order transformations. The network can be trained on pairs of point clouds before and after deforming with noisy scene flow.

The multibody equivariance is approximate because of 1.) descretization in EPN, 2.) the global backbone can’t handle deformation.

**Strengths:**

- The paper exploits strong geometry inductive bias — Equivariance, into the segmentation problem, which should be highlighted and emphasized more to the community.
- The system is tested across different settings from articulated objects to outdoor scenes and the experiments are convincing.

**Weaknesses:**

- The main weakness of this paper is that the “equivariance” is not exact, i.e. because there are pooling and information aggregation in the point cloud feature backbone when the target is deformed (not necessarily non-rigid, but rigid transformed, articulated, this is a word from equivariant learning theory), the overall point cloud changes in such cases, but the information can be propagated across rigid parts so the features of each point do not transform purely following its parts transformation. This is because without knowing the mask, the group action of SE(3)xSE(3)x…xSE(3) on the point cloud can not be defined.
- Another weakness of this paper is that the system should take in at least a pair of point as input during inference (as far as the reviewer currently understand, please correct him if this is wrong). So the task is relatively easy because by comparing two point clouds (even by brutal-force enumerating and classical geometric algorithms ) the segmentation of rigid moved parts may be trivially found in most cases. And the system seems only to focus on moving parts rather than semantic or instance parts that may not be moved.
- The reviewer is also curious why the author chooses to use EPN as the backbone, is there a specific reason for such a design? Note EPN is only approximate equivariance due to the discretization. In other words, will such a framework still work for other backbones like TFN or VNN?

**Questions:**

See weakness.

**Limitations:**

The limitation is explicitly highlighted in the paper, but the reviewer hasn't found the social impact claim.

---

> ### Author Rebuttal · Authors · 2023-08-10
>
> We appreciate your thorough feedback and are grateful for your recognition of our exploitation of applying equivariance to segmentation and our convincing experiments. We have carefully considered each of your concerns and have provided detailed responses to each one below.
>
> **Q1:** The “equivariance” is not exact, i.e. because there are pooling and information aggregation in the point cloud feature backbone when the target is deformed (not necessarily non-rigid, but rigid transformed, articulated, this is a word from equivariant learning theory), the overall point cloud changes in such cases, but the information can be propagated across rigid parts so the features of each point do not transform purely following its parts transformation. This is because without knowing the mask, the group action of SE(3)xSE(3)x…xSE(3) on the point cloud can not be defined.
>
> **A1:** Thank you for helping us improve our work. We agree that per-point features cannot transform strictly along with part-level transformation, resulting in an inexactly defined local "equivariance". However, the inexact local equivariance is proven to be sufficiently effective in practice from existing work [73]. Specifically, EON [73] utilizes object-level equivalence in fully-supervised object detection, of which the Region Context Aggregation is not perfectly equivariant. In this paper, we explore the feasibility of part-level equivariance in unsupervised multi-body segmentation, and it performs well on various datasets. We appreciate your insight and will include a discussion on the inexactness of equivariance to Sec 5 (Limitations).
>
> **Q2:** Another weakness of this paper is that the system should take in at least a pair of point as input during inference (as far as the reviewer currently understand, please correct him if this is wrong). So the task is relatively easy because by comparing two point clouds (even by brutal-force enumerating and classical geometric algorithms) the segmentation of rigid moved parts may be trivially found in most cases. And the system seems only to focus on moving parts rather than semantic or instance parts that may not be moved.
>
> **A2:** Thank you for your valuable comment. We would like to clarify that our model is capable of taking a single frame as input **during inference** and can segment parts that may not be moved. In the training stage, our model indeed requires at least two frames since scene flow serves as a supervisory signal in the unlabeled setting. However, the segmentation head can work without the help of motions during inference. In the experiments, static cars/trucks parked at the side of the road frequently occur on KITTI-SF (Table 4 in the main body of this paper), KITTI-Det & SemanticKITTI (**A3** in our response to Reviewer ZttT), and our model still performs comparably or better than state-of-the-art methods. We will include additional notes in Figure 2 to provide further clarification on the input data.
>
> **Q3:** The reviewer is also curious why the author chooses to use EPN as the backbone, is there a specific reason for such a design? Note EPN is only approximate equivariance due to the discretization. In other words, will such a framework still work for other backbones like TFN or VNN?
>
> **A3:** (1) The choice of using EPN as the backbone in our paper was made based on its robustness (Lines 121 - 125). By regressing the residual after selecting the rotational anchor, instead of directly regressing the entire rotation, vanilla EPN exhibits robust global equivariance, which is adopted in some existing unsupervised tasks like 6D pose estimation [43]. Designed for unsupervised learning, our model also adopts EPN as the backbone. (2) Theoretically, our framework should still work with other backbones such as TFN or VNN, as long as its per-point equivariant feature is accessible. Experimentally, it would require further investigation to determine their robustness to the part-level uncertainty and the unsupervised setting, which would be beyond the scope of this paper. We appreciate your suggestion and will make sure such explorations are included in future work.

---

> > ### Comment · Reviewer_S5HY · 2023-08-19
> >
> > The authors' responses address most of my concerns, and after reading other reviews I lean to keep my positive rating.

---

> > > ### Author Response · Authors · 2023-08-19
> > > **Response to Reviewer S5HY**
> > >
> > > Thank you for your review and for taking the time to read our responses. We are glad to hear that our responses have addressed most of your concerns and that you have decided to keep your positive rating. We really appreciate your feedback :)

---

### Official Review · Reviewer_ZttT · 2023-07-06

**Soundness:** 3 good
**Presentation:** 3 good
**Contribution:** 3 good
**Rating:** 6
**Confidence:** 4

**Summary:**

This paper proposes a unified framework to simultaneously handle rigid segmentation and motion estimation. Specifically, an probabilistic SE(3)-equivariant network is presented to obtain transformation-invariant features, which is followed by two heads for rigid segmentation and motion estimation. Due to the interdependent nature between the above two tasks, the authors utilize a unified training strategy to jointly optimize the two heads. Extensive experiments on four datasets show that this method can yield good performance with low computational complexity.


**Strengths:**

-Significance and originality

Unsupervised rigid segmentation and motion estimation are challenging. Taking the generalization requirement into account, the authors introduce a probabilistic part-level SE(3)-equivariant feature encoder to capture transformation-invariant features, which is different from the global SE(3) equivariance. And it is simple but makes sense.
A unified training strategy to jointly optimize the outputs of two heads in an online fashion is well motivated. Although many other tasks also employ joint optimization, this is the first one for this task.

-Clarity

The paper contains enough details and definitions of the proposed contributions.

-Experiment

Quantitative results show a clear improvement compared to previous methods.


**Weaknesses:**

-Time cost of the proposed methods and other methods.

-The author should systematically analyze the robustness on initial scene flow. Does the performance drop a lot with poor initial scene flow results? (add some noise or with insufficiently trained scene flow results).

-This paper relies on rigidity constraint and works well on finite datasets. The SAPIEN and OGC-DR datasets are without background and well-separated. The authors can conduct some experiments on large outdoor scenes, which are more complicated.


**Questions:**

please see the weakness.

**Limitations:**

-In related works, deep learning on dynamic point clouds is too highly summarized. The authors can present more details.

-The authors can visualize more segmentation results and intermidiate results so that the proposed method can be better understood.

---

> ### Author Rebuttal · Authors · 2023-08-10
>
> Thank you for your constructive feedback. We are grateful for your recognition of the significance and originality of our training strategy based on part-level SE(3)-equivariance, clarity, and experiments. We have carefully reviewed each of your concerns and all the additional experiments would be added to the supplementary materials accordingly.
>
> **Q1:** Time cost of the proposed methods and other methods.
>
> **A1:** We conduct experiments to compare the training and the testing time costs on the whole dataset of SAPIEN, reported as follows:
>
> | Model | Fully-supervised Training (Hours) | Unsupervised Training (Hours) | Inference (Hours) |
> |-------|--------------------|-----------------------|-----------|
> | MBS   |        18.7         |          ---         |    3.2      |
> | OGC   |        4.9         |          21.8         |    0.6   |
> | Ours  |        4.5         |          17.3         |    0.8  |
>
> Our inference time is slightly longer than that of the OGC model due to the additional processing time required by the EPN to handle the 60-angle icosahedral group. However, our model has a strong ability to capture rigid information, which allows it to converge with fewer training iterations and results in a shorter overall training time.
>
> **Q2:** Does the performance drop a lot with poor initial scene flow results? (add some noise or with insufficiently trained scene flow results).
>
> **A2:** We conduct additional experiments to analyze the influence of initial scene flow on the performance of segmentation. By adding various intensities of zero-centered Gaussian noise to scene flow, the segmentation result on SAPIEN is reported as follows:
>
> |  Variance of Gaussian Noise   | AP$\uparrow$ | PQ$\uparrow$ | F1$\uparrow$ | Pre$\uparrow$ | Rec$\uparrow$ | mIoU$\uparrow$ | RI$\uparrow$ |
> |----------------------------|--------------|--------------|--------------|---------------|---------------|----------------|--------------|
> |0 (w/o Noise)|63.8|61.3|77.3|84.2|71.3|63.7|75.4|
> |0.01|62.7|60.1|76.2|82.0|71.1|63.7|75.1|
> |0.025|60.3|57.4|73.0|75.6|70.6|62.6|74.8|
> |0.05|53.8|32.9|46.5|36.5|64.2|56.7|68.1|
> |0.1|49.2|27.1|40.3|30.2|60.5|53.4|65.2|
>
> Our model has been observed to be robust to noisy scene flow within a variance range of 0.01 to 0.025. However, as the intensity of the noise increases to 0.05, there is a noticeable decline in performance.
>
> **Our response to your Q3-Q5 will be uploaded once the function of interactive comment is available.**

---

> > ### Author Response · Authors · 2023-08-10
> > **Response to Reviewer ZttT (Q3)**
> >
> > **Q3:** This paper relies on rigidity constraint and works well on finite datasets. The SAPIEN and OGC-DR datasets are without background and well-separated. The authors can conduct some experiments on large outdoor scenes, which are more complicated.
> >
> > **A3:** Thank you for this insightful suggestion! Following the same evaluation protocol of OGC [61], we further evaluate the generalizability of our model from KITTI-SF to two larger outdoor datasets, i.e., KITTI-Det [1*] and SemanticKITTI [2*], and compare the performance with the results reported in OGC:
> >
> > (1) KITTI-Det
> >
> > |                            | AP$\uparrow$ | PQ$\uparrow$ | F1$\uparrow$ | Pre$\uparrow$ | Rec$\uparrow$ | mIoU$\uparrow$ | RI$\uparrow$ |
> > |----------------------------|--------------|--------------|--------------|---------------|---------------|----------------|--------------|
> > | OGC$_{sup}$                | 51.4         | 41.0         | 49.1         | 43.7          | 56.0          | 66.2           | 91.0         |
> > | Ours$_{sup}$                | **52.5**         | **43.3**         | **51.8**         | **47.5**          | **57.0**          | **68.0**           | **92.6**         |
> > | | | | | | | | |
> > | OGC$_{unsup}$                | 40.5         | 30.9         | 37.0         | 30.8          | **46.5**          | **60.6**           | 86.4         |
> > | Ours$_{unsup}$                | **41.3**         | **32.9**         | **38.8**         | **35.3**          | 43.1          | 60.2           | **87.2**         |
> >
> > (2) SemanticKITTI
> >
> > | Sequences               | Methods        | AP$\uparrow$ | PQ$\uparrow$ | F1$\uparrow$ | Pre$\uparrow$ | Rec$\uparrow$ | mIoU$\uparrow$ | RI$\uparrow$ |
> > |-------------------------|----------------|--------------|--------------|--------------|---------------|---------------|----------------|--------------|
> > | 00 - 10              | OGC$_{sup}$    | 53.8         | 41.3         | 48.1         | 40.1          | 60.0          | 68.3           | 90.0         |
> > |                         | Ours$_{sup}$   | **60.1**         | **47.6**         | **55.4**         | **48.6**          | **64.4**          | **71.9**           | **93.4**         |
> > |                         |                |              |              |              |               |               |                |              |
> > |                         | OGC$_{unsup}$  | 42.6         | 30.2         | 35.3         | 28.2          | 47.3          | 60.3           | 86.0         |
> > |                         | Ours$_{unsup}$ | **46.9**         | **31.6**         | **36.9**         | **29.0**          | **50.6**          | **63.2**           | **88.7**         |
> > |                         |                |              |              |              |               |               |                |              |
> > | 00 - 07 & 09 - 10 | OGC$_{sup}$    | 55.3         | 41.8         | 48.4         | 40.1          | 61.1          | 69.9           | 90.3         |
> > |                         | Ours$_{sup}$   | **60.5**         | **48.1**         | **55.6**         | **48.8**          | **64.7**          | **73.2**           | **93.8**         |
> > |                         |                |              |              |              |               |               |                |              |
> > |                         | OGC$_{unsup}$  | 43.6         | 30.5         | 35.5         | 28.1          | 48.2          | 62.1           | 86.3         |
> > |                         | Ours$_{unsup}$ | **47.4**         | **31.7**         | **36.8**         | **28.7**          | **51.0**          | **64.8**           | **89.3**         |
> > |                         |                |              |              |              |               |               |                |              |
> > | 08                      | OGC$_{sup}$    | 49.4         | 39.2         | 46.6         | 40.0          | 55.8          | 60.3           | 88.3         |
> > |                         | Ours$_{sup}$   | **58.4**         | **46.0**         | **54.4**         | **47.8**          | **63.1**          | **65.8**           | **91.7**         |
> > |                         |                |              |              |              |               |               |                |              |
> > |                         | OGC$_{unsup}$  | 38.6         | 29.1         | 34.7         | 28.6          | 44.0          | 51.8           | 84.3         |
> > |                         | Ours$_{unsup}$ | **44.2**         | **31.0**         | **37.3**         | **30.0**          | **49.1**          | **55.8**           | **86.1**         |

---

> > > ### Author Response · Authors · 2023-08-10
> > > **Response to Reviewer ZttT (Q3-Q5)**
> > >
> > > Unlike the stereo-based point clouds in KITTI-SF, the point clouds in the SemanticKITTI and KITTI-Det datasets are collected using LiDAR sensors, resulting in sparser data. In these **large-scale** and **low-density** outdoor settings, our model significantly outperforms the state-of-the-art approach OGC across all metrics on SemanticKITTI and on a majority of metrics on KITTI-Det. In addition to the above experiments, we plan to include more results and comparisons (e.g., training our model from scratch on these large-scale outdoor scenes) in our revised manuscript.
> > >
> > > [1*] Geiger, Andreas, Philip Lenz, and Raquel Urtasun. "Are we ready for autonomous driving? the kitti vision benchmark suite." 2012 IEEE conference on computer vision and pattern recognition. IEEE, 2012.
> > >
> > > [2*] Behley, Jens, et al. "Semantickitti: A dataset for semantic scene understanding of lidar sequences." Proceedings of the IEEE/CVF international conference on computer vision. 2019.
> > >
> > > **Q4:** In related works, deep learning on dynamic point clouds is too highly summarized. The authors can present more details.
> > >
> > > **A4:** We appreciate your suggestion to provide more details in the related works section of our manuscript. We will revise Sec 2 to provide a more comprehensive overview of the existing literature on deep learning on dynamic point clouds. Thank you for bringing this to our attention. If you have any specific citations or references that you would like us to include, please let us know and we will be happy to incorporate them into our revised manuscript.
> > >
> > > **Q5:** The authors can visualize more results so that the proposed method can be better understood.
> > >
> > > **A5:** We have provided additional visualizations, which are attached as a PDF in the global response, across four datasets: OGC-DR, OGC-DSRV, SAPIEN, and KITTI-SF with multiple rigid parts. We have also included some failure cases in challenging scenes on the SAPIEN dataset, as indicated by the green circles on the final two rows of the SAPIEN visualization. For more details on the visualizations, please refer to the global response.

---

> > > > ### Comment · Reviewer_ZttT · 2023-08-20
> > > >
> > > > I thank the authors for their response. The response have addressed my concerns. I keep my original rating.

---

> > > > > ### Author Response · Authors · 2023-08-21
> > > > > **Response to Reviewer ZttT**
> > > > >
> > > > > We are very grateful for your reply and your positive feedback. We thank you for your careful and thorough review of our paper. We are happy to hear that our response have addressed your concerns and that you keep your original rating. We hope that our paper can make a significant contribution to the literature and that you will enjoy reading it. Thank you again for your kind words and support.

---

### Official Review · Reviewer_mTMF · 2023-07-07

**Soundness:** 3 good
**Presentation:** 3 good
**Contribution:** 3 good
**Rating:** 6
**Confidence:** 2

**Summary:**

This paper proposes a part-level SE(3)-equivariant framework for multi-body rigid motion modeling. The two heads that take in the SE(3) features, segmentation and motion estimation, are carefully crafted based on their inherit invariant and equivariant characteristics. The relationship of scene flow, rigid segmentation, and multi-body transformation is then exploited to derive an unsupervised optimization strategy. It achieves state-of-the-art in multiple datasets while significantly reducing the parameters and training computations required.

**Strengths:**

The structure of this paper is very clear, providing a comprehensive explanation of the research problem, motivation, background, principles, methods, and experiments. The text and figures have been meticulously polished.

This work effectively utilizes SE(3) equivariance in rigid segmentation and multi-body motion estimation, and is outstanding in terms of experimental results and model size.

**Weaknesses:**

Rigid segmentation and multi-body motion estimation are intrinsically coupled. I wonder if this coupling might also have negative effects on the model, for instance, inaccurate segmentation leading to poorer scene flow estimation results.

**Questions:**

Please refer to the weaknesses section, refute views or answer questions or improve the paper.

**Limitations:**

The limitations has been discussed.

---

> ### Author Rebuttal · Authors · 2023-08-10
>
> We appreciate your valuable feedback and are grateful for your recognition of our writing and contributions presented in our paper, such as the utilization of SE(3) equivariance, experiments, and model size. We have carefully considered your main concern and have provided detailed responses below.
>
> **Q1:** Rigid segmentation and multi-body motion estimation are intrinsically coupled. I wonder if this coupling might also have negative effects on the model, for instance, inaccurate segmentation leading to poorer scene flow estimation results.
>
> **A1:** We agree that rigid segmentation and multi-body motion estimation are intrinsically coupled, and this coupling can have both positive and negative effects on the model.
>
> (1) It is natural that inaccurate segmentation would lead to poor scene flow estimation. To further explore the influence of inaccurate segmentation, we conduct additional experiments on the validation set of SAPIEN. By adding different levels of noise to the predictive segmentation, we compare the performance change in unsupervised motion estimation:
>
> | Noise Level | EPE3D$\downarrow$ | AccS$\uparrow$ | AccR$\uparrow$ | Outl$\downarrow$ |
> |-------------|-------------------|----------------|----------------|------------------|
> | 5%          | 4.34              | 15.0           | 32.2           | 84.1             |
> | 15%         | 6.79              | 11.6           | 26.1           | 87.6             |
> | 25%         | 9.29              | 9.28           | 21.5           | 89.9             |
>
> It is observed that poor segmentation would lead to low performance on scene flow estimation results.
>
> (2) Experiments also demonstrate that poor scene flow could result in performance degradation in rigid segmentation. Please refer to **A2** in our response to Reviewer ZttT for more details.
>
> (3) To mitigate the negative effect of this coupling, this paper proposes some corrective schemes, such as consensus $\beta^{s(i)}_{kl}$ (Lines 205 - 210) and smoothly updating scene flow (Lines 194 - 200). In the unsupervised setting, the benefits of coupling rigid segmentation and multi-body motion estimation outweigh the potential drawbacks. Experiments show that our model is able to leverage the complementary information provided by each task to improve its performance on both.
>
> For future work, we encourage the community to improve our model to alleviate the side effect of the coupling under the unsupervised setting. Thank you again for your insightful comment.

---

> > ### Comment · Reviewer_mTMF · 2023-08-21
> >
> > Sorry for my late reply. The rebuttal can address my concerns. Referring to the opinions of other reviewers, I will retain my score.

---

> > > ### Author Response · Authors · 2023-08-21
> > > **Response to Reviewer mTMF**
> > >
> > > Thank you for your reply and your constructive feedback. We appreciate your time and effort in reviewing our paper. We are glad to hear that our rebuttal can address your concerns and that you will retain your score. We hope that our paper can contribute to the advancement of the field and that you will find it useful for your future research. Thank you again for your valuable comments and suggestions.

---

### Official Review · Reviewer_86bd · 2023-07-07

**Soundness:** 3 good
**Presentation:** 2 fair
**Contribution:** 3 good
**Rating:** 5
**Confidence:** 2

**Summary:**

Modeling multi-body rigid movements involves - rigid segmentation which is category-agnostic and motion estimation which is open-set. This work proposes a part-level SE(3) equivariant network that takes in point-clouds corresponding to video frames and can effectively estimate category-agnostic open-set motion.


**Strengths:**

The work leverages the point level feature equivariance from existing works and uses it to obtain invariant representations for the rigid segmentation and then further uses it to estimate motion using an equivariant motion head.

The approach is able to provide state of the art performance with much lesser flops.

It makes note of the interdependence between the tasks of rigid segmentation and motion estimation and uses that interdependence for unsupervised training.

**Weaknesses:**

The paper writing could be better so that the high-level idea is more easily understood. The contributions of the work could be summarized in fewer words in the introduction section making the work much easier to follow.

SE(3) has been used multiple times in the paper including the title but it has not been described in the paper anywhere.



**Questions:**

Please address the issues mentioned in the weakness section

**Limitations:**

The authors have discussed limitations of their proposed approach.

---

> ### Author Rebuttal · Authors · 2023-08-10
>
> Thank you for your valuable feedback. We appreciate your recognition of our improvements on feature equivariance, the high performance and low FLOPs, and our use of interdependence for unsupervised training. Please find our responses to each concern below.
>
> **Q1:** The paper writing could be better so that the high-level idea is more easily understood. The contributions of the work could be summarized in fewer words in the introduction section making the work much easier to follow.
>
> **A1:** We really appreciate your advice to improve the readability of this paper. Currently, the main summary for each contribution is highlighted in italic format. We realize that this format may be hard to read, and will reformat the section by describing the details first and having shorter bullet points towards the end of Introduction.
>
> **Q2:** SE(3) has been used multiple times in the paper including the title but it has not been described in the paper anywhere.
>
> **A2:** Thank you for bringing this to our attention and we apologize for not including a detailed description of SE(3) in our paper. SE(3) stands for the **S**pecial **E**uclidean group in three dimensions (**3**D), which is the group of rigid body transformations in three-dimensional space. We will make sure to include a definition and explanation of SE(3) in Sec 3.1.

---

### Official Review · Reviewer_sjc7 · 2023-07-22

**Soundness:** 2 fair
**Presentation:** 2 fair
**Contribution:** 3 good
**Rating:** 6
**Confidence:** 3

**Summary:**

This paper presents a method that estimates the part segmentation, and the rigid motions of the parts, for an articulated object (with multiple rigid parts), given as input a sequence of point clouds of that object. The method works by first computing SE3-equivariant features for the points, and then processing with an SE3-invariant head for segmentation estimation (pooling across rotation groups), and then processing with an SE3-equivariant head for rigid motion estimation (pooling across points within estimated segments). The method also initializes a scene flow estimate from a pre-trained off-the-shelf model, and then iteratively updates this to match the estimated rigid motion. Finally, the model uses the weighted-Kabsch algorithm to estimate transformations for the parts according to the flow and segmentation, which yields optimization targets that encourage the estimates to be self-consistent.

**Strengths:**

This paper makes a well-motivated case for using a combination of equivariant and invariant features to jointly solve rigid part segmentation and motion estimation. The proposed network has fewer parameters and runs faster than the baselines, and also achieves higher accuracy, in both the supervised setting and the unsupervised setting. Jointly optimizing the motion, segmentation, and scene flow, makes sense.

**Weaknesses:**

I found this paper very difficult to follow. I am quite sure that the content in the main text is not a complete description of a reproducible method. Nearly every section defers to the supplementary material for additional information. In my reading of the guidelines, the supplementary material is for supporting the paper, not for describing key details of the method. The method in the paper should make sense on its own.

 Because my main complaint is on clarity, I'll put a lot of questions, but please interpret some of the questions as reflecting weaknesses, because these are things that should be clear in the work.

**Questions:**

Do the two point clouds correspond? I suppose not, because then motion can be computed just from taking a difference of the positions.


The problem statement in 3.2 suggests a sequence, but later (e.g., in 3.3) it seems like the sequence length might just be 2. Is the sequence length ever not 2?


The w(p^i_k,g_j) used in equation 2 are described as "derived through a 1x1 convolution". What does this mean exactly?

The paper talks about "fusing such invariant representations u^i_{k(1)},... from h layers" -- how is this fusing done exactly?

The paper says "the segmentation head outputs a probabilistic prediction M^k of the rigid mask" -- what makes it probabilistic exactly? This idea of "probabilistic" masks is repeated many times through the paper, and at some point there are also "probabilistic SE3 features" (actually only in Figure 3). In my experience, the word "probabilistic" is usually used to indicate something non-deterministic, but in this case it seems like it might actually be deterministic. What exactly is meant by a probabilistic mask or probabilistic feature?

The paper says "the motion head is supposed to *handle these uncertain category-agnostic parts*" (italics in the paper). Why are those words in italics? What is meant by "handle" here? It seems like a special usage of the word.

Equation 3 looks like it might just be depicting an element-wise product. Is this correct? It would be great to write this more efficiently.

The paper says "the specific category labels can be agnostic to the model". What does it mean for category labels to be agnostic to a model?

The paper says "Based on the correlated feature Ckl, the motion head estimates rotation R^s_{kl} and translation t^s_{kl} of each rigid part s." This is a critical part of the model. How are the correlations used exactly, what is the predictor, how are the rotation and translation outputs represented, and how are the outputs supervised?

Under equation 6, it says "In this manner, improved scene flow is capable of providing enhanced supervision to learn segmentation masks and motion estimates." I don't see how this follows from the equation. The equation only produces an updated scene flow, using a convex combination of the previous estimate and the rigid estimate. I see no "supervision" (or "enhanced supervision") applied to segmentation masks here.

In the final paragraph of the method, it says "the motion head is optimized by the rotation component of T^s_{kl}, and estimates corresponding translation by minimizing our probabilistic part-level distance." What does it mean to optimize the motion head by rotation? What does it mean to estimate translation by minimizing "probabilistic part-level distance"? It seems like these are critical details in the procedure.

**Limitations:**

Looks OK

---

> ### Author Rebuttal · Authors · 2023-08-10
>
> Thank you for your valuable feedback! We are grateful for your acknowledgment of the motivation, novelty, and contributions of our model in terms of its parameters, complexity, and accuracy. While we understand your concern regarding the clarity of certain sections, we would like to address each of your concerns in detail below and assert that these corrections can indeed be addressed for the camera-ready version. We hope our answers, along with the support of Reviewer mTMF and ZttT that the explanation of the paper is "clear" and "detailed", can alleviate your concerns and give more merit based on the impact and originality of our method.
>
> **Q1:** Do the two point clouds correspond? I suppose not, because then motion can be computed just from taking a difference of the positions.
>
> **A1:** No, we do not assume that these point clouds correspond to one another. The correspondence between points in the two frames is unknown.
>
> **Q2:** The problem statement in 3.2 suggests a sequence, but later (e.g., in 3.3) it seems like the sequence length might just be 2. Is the sequence length ever not 2?
>
> **A2:** The length of an input sequence can exceed 2. For simplicity, we only describe how to process a pair of point clouds, but the method can be extended to a sequence longer than two frames, following common practice. In accordance with previous works [32, 61, 68], the sequence is input into a model as a pair of either any two frames (for the SAPIEN dataset) or consecutive frames (for the OGC dataset) at each iteration. We will further describe this at the start of 3.3 in the revision.
>
> **Q3:** The $w(p^i_k,g_j)$ used in equation 2 are described as "derived through a 1x1 convolution". What does this mean exactly?
>
> **A3:** This notation follows the paper of the original EPN [9]: the convolution of the $j^{th}$ icosahedral angle $g_j$ takes a point $p^i_k$ as its input and outputs the selection probability $w$. In other words, the value of $w$ is obtained from a 1x1 convolution, which is implemented through a point-wise mapping of the features from the last layer for a dimensional change.
>
> **Q4:** The paper talks about "fusing such invariant representations $u^i_{k(1)},...$ from $h$ layers" -- how is this fusing done exactly?
>
> **A4:** We apologize for not including a detailed description of our fusing process in the main body of the paper. Due to page limitations, our initial approach was to explain the high-level end-to-end idea of our method and have conformed to putting the details in the original supplementary material. Specifically, the fusing process is illustrated in Figure 2 of Supp. The multi-layer invariant representations are fused through an hourglass structure, where we downsample the features with Set Abstraction (SA) and then upsample with Feature Propagation (FP) modules [55]. Subsequently, this is followed by an attention-based decoder, as practiced in OGC [61]. We will make sure to include a brief description in Sec 3.3 in the main body to make it more readable.
>
> **Q5:** The paper says "the segmentation head outputs a probabilistic prediction $M^k$ of the rigid mask" -- what makes it probabilistic exactly? This idea of "probabilistic" masks is repeated many times through the paper, and at some point there are also "probabilistic SE(3) features" (actually only in Figure 3). In my experience, the word "probabilistic" is usually used to indicate something non-deterministic, but in this case it seems like it might actually be deterministic. What exactly is meant by a probabilistic mask or probabilistic feature?
>
> **A5:** We use the term “probabilistic” to refer to the fact that the segmentation head outputs a prediction of the rigid mask in the form of a probability distribution. Instead of a "hard" binary rigid partition, the mask represents the probability of each point “softly” belonging to different rigidities. Similarly, the term “probabilistic SE(3) features” refers to features that correspond to a probability distribution over SE(3) transformations, rather than a fixed rigid transformation. We apologize for any confusion and will change this term to  **soft prediction** in Sec 3.3.
>
> **Q6:** The paper says "the motion head is supposed to _handle these uncertain category-agnostic parts_" (italics in the paper). Why are those words in italics? What is meant by "handle" here? It seems like a special usage of the word.
>
> **A6:** We use italics to emphasize that the motion head can deal with parts of the data that are uncertain (estimated from the segmentation head, without ground-truth partition) and do not belong to a specific category. By “handle”, we mean that the motion head is designed to process these uncertain parts in order to produce a meaningful output.
>
> **Q7:** Equation 3 looks like it might just be depicting an element-wise product. Is this correct? It would be great to write this more efficiently.
>
> **A7:** Thank you for the question. The symbol $\cdot$ means a broadcasting element-wise product over the $D$-dimensional feature channel, which is not exactly the same as a standard element-wise operation. Since $\hat{m}_k^i$ and $\theta (p_k^i,g_j)$ have different sizes, a standard element-wise operation cannot be directly applied.
>
> **Q8:** The paper says "the specific category labels can be agnostic to the model". What does it mean for category labels to be agnostic to a model?
>
> **A8:** This sentence occurs on Line 181, which means that our model does not require any category labels of the parts (e.g., 'Lense', 'Temple', 'Nose Pad' for eyeglasses), and the ground truth of part-level correspondence between frames is unknown to this model. We will change the wording to convey this idea more clearly.
>
> **Our response to your Q9-Q11 will be uploaded once the function of interactive comment is available.**

---

> > ### Author Response · Authors · 2023-08-10
> > **Response to Reviewer sjc7 (Q9-Q11)**
> >
> > **Q9:** The paper says "Based on the correlated feature $C_{kl}$, the motion head estimates rotation $R^s_{kl}$ and translation $t^s_{kl}$ of each rigid part $s$." This is a critical part of the model. How are the correlations used exactly, what is the predictor, how are the rotation and translation outputs represented, and how are the outputs supervised?
> >
> > **A9:** Originally, due to the page limitation and its close similarity following the approach of EPN [9], we included the details in the supplementary material (Lines 38-46). We realize that this may cause slight ambiguity and will briefly describe this method towards the end of Line 183 in the main paper. The summary is as follows. The predictor of rotation is based on a 1×1 convolution. The anchor $g_{kl}^s$ is chosen from $\mathcal{G}$ by minimizing the registration error and then optimizing the residual $r_{kl}^{s}$. The rotation is computed as $\hat{\mathbf{R}}^s_{kl}=g_{kl}^{s}r_{kl}^{s}$. The translation is derived from the minimal weighted distance between the transformed frame of $P_k$ and the origin $P_l$, where $\hat{\mathbf{t}}^s_{kl}=argmin_{\mathbf{t}} d(\hat{\mathbf{R}}^s_{kl}P_k + \mathbf{t}, P_l)$ and $d$ is the chamfer loss weighted by the mask predictions.
> >
> > **Q10:** Under equation 6, it says "In this manner, improved scene flow is capable of providing enhanced supervision to learn segmentation masks and motion estimates." I don't see how this follows from the equation. The equation only produces an updated scene flow, using a convex combination of the previous estimate and the rigid estimate. I see no "supervision" (or "enhanced supervision") applied to segmentation masks here.
> >
> > **A10:** Thank you for bringing this to our attention. We apologize for any confusion caused by the sentence in question. Instead of only continuing from the previous text (Equation 6), this sentence mainly serves as a transition to the next two paragraphs (Lines 201-216), which subsequently describe how the improved scene flow supervises the segmentation (Lines 201-211) and motion estimation (Lines 212-216).
> >
> > **Q11:** In the final paragraph of the method, it says "the motion head is optimized by the rotation component of $T^s_{kl}$, and estimates corresponding translation by minimizing our probabilistic part-level distance." What does it mean to optimize the motion head by rotation? What does it mean to estimate translation by minimizing "probabilistic part-level distance"? It seems like these are critical details in the procedure.
> >
> > **A11:** Based on the original supplementary material (Lines 54-60), we create a summary that will be added to the revised paper (we will make our contributions in Introduction more concise to compensate for the space needed). The summary is as follows. Based on the prediction of segmentation masks, the part-level rigid transformation can be computed by the weighted-Kabsch algorithm [32, 27]. By minimizing a mean SO(3) loss (described in the original paper of EPN [9]) from the rotation component of this rigid transformation, the estimated rotation ($\hat{\mathbf{R}}^s_{kl}$ in **A9** ) is optimized in a differentiable manner. The corresponding translation is then estimated by minimizing the part-level chamfer distance (through the formula of $\hat{\mathbf{t}}^s_{kl}$ described in **A9**), which aims to find the translation that best aligns the two point clouds.
> >
> > We will make sure to revise our paper to ensure that the method is fully described and further understandable. We appreciate your questions and will address them in detail to clarify any weaknesses or misunderstandings in the revised manuscript. Thank you again for bringing these issues to our attention.

---

### Author Rebuttal · Authors · 2023-08-10

We thank all reviewers for their valuable comments and suggestions. We would like to highlight some strengths of our paper pointed out by reviewers, including a **well-motivated** (sjc7) and **original** (Ztt7) idea, **extensive and convincing experiments with impressive results surpassing state-of-the-art methods** (86bd, Ztt7, S5HY, XLXX), and **clear and detailed explanation of the motivation, contributions, method, and experiments** (mTMF, ZttT). We provide individual feedback for every review to address specific concerns.

We also provide additional visualizations in the attached pdf file. The file includes (a) challenging scenes (multiple parts) on SAPIEN, with successful and failure cases (errors as circled in green on the final two rows); (b) visualizations on **all of the other datasets** (OGC-DR, OGC-DRSV, and KITTI-SF). All of the results ("Ours") are obtained from our model in the **unsupervised setting**, while "GT" means the ground-truth segmentation.

---

### Decision · Program_Chairs · 2023-09-21

**Decision:**

Accept (poster)

**Comment:**

This paper presents a method that estimates the part segmentation and the rigid motions of the parts, for an articulated object (with multiple rigid parts), given as input a sequence of point clouds of that object. All reviewers agree on the extensive experiments and the good empirical performance. Rsjc7 and R86bd raised concerns regarding clarity and the method being not self-contained. The rebuttal submitted by the authors provided additional experiments and visualizations and addressed to the raised concerns. The authors are encouraged to make the paper self-contained and rigorously describe their method to make it accessible to a wide audience.